

# Can sampling biases explain the discrepancies between lower stratospheric water vapour trend estimates derived from the FPH observations at Boulder and a merged zonal mean satellite data set?

Stefan Lossow[1], Dale F. Hurst[2], Karen H. Rosenlof[2], Gabriele P. Stiller[1], Thomas von Clarmann[1], Sabine Brinkop[3], Martin Dameris[3], Patrick Jöckel[3], Doug E. Kinnison[4], Johannes Plieninger[1], David A. Plummer[5], Felix Ploeger[6], William G. Read[7], Ellis E. Remsberg[8], James M. Russell[9], and Mengchu Tao[6]

[1]Karlsruhe Institute of Technology, Institute for Meteorology and Climate Research, Hermann-von-Helmholtz-Platz 1, 76344 Leopoldshafen, Germany.
[2]NOAA Earth System Research Laboratory, Global Monitoring Division, 325 Broadway, Boulder, CO 80305, USA.
[3]Deutsches Zentrum für Luft- und Raumfahrt (DLR), Institut für Physik der Atmosphäre, 82234 Oberpfaffenhofen-Wessling, Germany.
[4]University of Colorado, Atmospheric Chemistry Observations & Modeling Laboratory, P.O. Box 3000, Boulder, CO 80305-3000, USA.
[5]Environment and Climate Change Canada, Climate Research Branch, 550 Sherbrooke ouest, Montréal, Quebec H3A 1B9, Canada.
[6]Forschungszentrum Jülich, Institute for Energy and Climate Research: Stratosphere (IEK–7), Leo-Brandt-Straße, 52425 Jülich, Germany.
[7]Jet Propulsion Laboratory, 4800 Oak Grove Drive, Pasadena, CA 91109, USA.
[8]NASA Langley Research Center, 21 Langley Boulevard, Hampton, VA 23681, USA.
[9]Hampton University, Center for Atmospheric Sciences, 23 Tyler Street Hampton, VA 23668, USA.

*Correspondence to:* Stefan Lossow (stefan.lossow@kit.edu)

**Abstract.** Trend estimates with different signs are reported in the literature for lower stratospheric water vapour considering the
time period between the late 1980s and 2010. The NOAA (National Oceanic and Atmospheric Administration) frost point hy-
grometer (FPH) observations at Boulder (Colorado, 40.0°N, 105.2°W) indicate positive trends (about $0.12\,\mathrm{ppmv \cdot decade^{-1}}$ –
$0.45\,\mathrm{ppmv \cdot decade^{-1}}$). Contrary, negative trends (approximately $-0.15\,\mathrm{ppmv \cdot decade^{-1}}$ – $-0.05\,\mathrm{ppmv \cdot decade^{-1}}$) are de-
rived from a merged zonal mean satellite data set for a latitude band around the Boulder latitude. Overall, the trend differ-
ences between the two data sets range from about $0.25\,\mathrm{ppmv \cdot decade^{-1}}$ to $0.45\,\mathrm{ppmv \cdot decade^{-1}}$, depending on altitude.
It has been proposed that a possible explanation for these discrepancies is a different temporal behaviour at Boulder and
the zonal mean, which simply indicates a sampling bias. In this work we investigate trend differences between Boulder and
the zonal mean using primarily simulations from ECHAM/MESSy (European Centre for Medium-Range Weather Forecasts
Hamburg/Modular Earth Submodel System) Atmospheric Chemistry (EMAC), WACCM (Whole Atmosphere Community Cli-
mate Model), CMAM (Canadian Middle Atmosphere Model) and CLaMS (Chemical Lagrangian Model of the Stratosphere).
On shorter time scales we address this aspect also based on satellite observations from UARS/HALOE (Upper Atmosphere



Research Satellite/Halogen Occultation Experiment), Envisat/MIPAS (Environmental Satellite/Michelson Interferometer for
Passive Atmospheric Sounding) and Aura/MLS (Microwave Limb Sounder). Overall, both the simulations and observations
exhibit trend differences between Boulder and the zonal mean. The differences are dependent on altitude and the time period
considered. The model simulations indicate only small trend differences between Boulder and the zonal mean for the time
period between the late 1980s and 2010. These are clearly not sufficient to explain the discrepancies between the trend esti-
mates derived from the FPH observations and the merged zonal mean satellite data set. Unless the simulations underrepresent
variability or the trend differences originate from smaller spatial and temporal scales than resolved by the model simulations,
trends at Boulder for this time period should be quite representative also for the zonal mean and even other latitude bands.
Trend differences for a decade of data are larger and need to be kept in mind when comparing results for Boulder and the zonal
mean on this time scale. Beyond that, we find that the trend estimates for the time period between the late 1980s and 2010 also
significantly differ among the simulations. They are larger than those derived from the merged satellite data set and smaller
than the trend estimates derived from the FPH observations.
**1    Introduction**
Water vapour in the stratosphere plays a fundamental role in the radiative budget and affects the ozone chemistry in this
atmospheric layer. In the lower stratosphere water vapour is the most important greenhouse gas. As such, it is part of an
important global warming feedback mechanism. A warmer climate increases lower stratospheric water vapour, leading to an
even warmer climate. Dessler et al. (2013) estimated this feedback to be $0.3\,\mathrm{W \cdot m^{-2}}$ for a temperature anomaly of $1\,\mathrm{K}$ at
$500\,\mathrm{hPa}$. Besides that, water vapour is a fundamental component of polar stratospheric clouds. The heterogeneous chemistry
on cloud particle surfaces is responsible for the severe ozone depletion in the lower stratosphere during winter and spring,
especially in the Antarctic (Solomon, 1999). Water vapour is also the main source of hydrogen radicals ($HO_x = OH, H, HO_2$)
in the stratosphere that contribute to ozone destruction through catalytic loss cycles (Brasseur and Solomon, 2005).

22        Thus, any change of stratospheric water vapour over a longer time scale has important implications (e.g. Dvortsov and

Solomon, 2001; Forster and Shine, 2002; Stenke and Grewe, 2005; Solomon et al., 2010; Riese et al., 2012; Maycock et al.,
2014; Gilford et al., 2016). In the past, the majority of studies related to longer term water vapour changes were based on
observations by the balloon-borne NOAA frost point hygrometer at Boulder (a more detailed description of the measurement
principle is provided in Sect. 2.2.4). These observations have been performed since 1980, typically once per month, providing
the longest time series of water vapour in the lower stratosphere. Positive trends over Boulder were first reported by Oltmans
and Hofmann (1995), then by Oltmans et al. (2000), Scherer et al. (2008) and finally Hurst et al. (2011). For the time period
from 1980 to 2010, Hurst et al. (2011) showed an overall increase of $0.24\,\mathrm{ppmv \cdot decade^{-1}}$ to $0.42\,\mathrm{ppmv \cdot decade^{-1}}$ for the
altitude range between $16\,\mathrm{km}$ and $26\,\mathrm{km}$ accompanied by significant variability on shorter time scales. 25% of the observed
increase could be associated to changes of methane (Hurst et al., 2011). The oxidation of this trace gas is the most important in
situ source of water vapour in the stratosphere. The other relevant source of water vapour in the stratosphere is transport from the
troposphere, which mainly occurs through the cold tropical tropopause region. One major pathway is slow ascent (accompanied





by large horizontal motions, Holton and Gettelman, 2001) where the amount of water vapour entering the stratosphere is mainly
controlled by the tropopause temperature (or better cold point temperature, Fueglistaler et al., 2009). Different changes of this
temperature have been reported. Rosenlof and Reid (2008) reported an overall negative trend for the time period from 1980
to 2003, which would correspondingly result in a decrease of lower stratospheric water vapour. Recent work by Randel et al.
(2017) indicates zero or slightly positive trends at the tropical tropopause for the time periods 1979 to 1997 and 1998 to 2014.
The other pathway thought to be of importance is the convective lofting of ice particles (Moyer et al., 1996; Dessler et al.,
2016; Avery et al., 2017). Once the particles reach the stratosphere, they evaporate and enhance the amount of stratospheric
water vapour. This process is not dependent on the (cold point) temperature. Balloon-borne observations indicated no trend
of the convective ice lofting into the stratosphere for the time period between 1991 and 2007 (Notholt et al., 2010). Based on
all these results it is difficult to assess what process(es) caused the 30 year net increase of lower stratospheric water vapour
observed by the FPH observations at Boulder (Hurst et al., 2011).
Satellite observations of stratospheric water vapour exist since 1978 (Gille and Russell, 1984), with some gaps. The instru-
ments have limited life times and thus individual data sets do not allow a trend analysis on the same time scale as the FPH
observations at Boulder. Recently, Hegglin et al. (2014) merged zonal mean data sets from seven satellite instruments. This
merging was achieved with help of CMAM simulations with specified dynamics (aka nudging) which acted as a transfer func-
tion. For each data set biases relative to the CMAM simulations were estimated. This assumes that the CMAM simulations pro-
vide a realistic representation of the water vapour variability (including trends) and that the satellite data sets do not have a drift
in the bias estimation period. With this bias information the individual data sets were then adjusted relative to the Aura/MLS
observations. Finally the average over all bias-corrected data sets was used for the merged data set. This data set covers the time
period between 1986 or 1988 (depending on latitude and altitude) and 2010, providing the opportunity to evaluate the trends
observed by the FPH observations at Boulder and to address water vapour changes on a more global scale. The trends derived
from the merged satellite data set for the zonal mean of the latitude around Boulder were negative below about $10\,\mathrm{hPa}$ and
positive above. This behaviour could be essentially also observed at all other latitudes. Below $20\,\mathrm{hPa}$ the percentage changes
up to 2010 were typically between $-5\%$ to $-10\%$, which roughly corresponds to a trend between $-0.05\,\mathrm{ppmv}\cdot\mathrm{decade}^{-1}$ and
$-0.15\,\mathrm{ppmv}\cdot\mathrm{decade}^{-1}$. Hegglin et al. (2014) attributed this trend to a reduced transport of water vapour into the stratosphere
as a consequence of lower tropopause temperatures and a changed circulation in the stratosphere. During the same period as
covered by the merged satellite data set, the FPH observations at Boulder still exhibit a clear increase of lower stratospheric
water vapour (Hurst et al., 2011).
Figure 1 provides a summary of the trend discrepancies between the FPH observations and the merged satellite data. The
trends derived from the merged satellite data set for the latitude band around Boulder (35°N – 45°N) are shown in green.
Below (above) the dashed line the satellite trends are representative for the time period from 1988 to 2010 (1986 to 2010).
The estimates are based on a digitisation of Fig. 5a in Hegglin et al. (2014). The extracted percentage trends were converted
to volume mixing ratio trends by assuming a constant reference mixing ratio of $3\,\mathrm{ppmv}$. Consequently, the trends presented in
Fig. 1 are approximations. They are likely an underestimate as the actual reference mixing ratio is probably somewhat larger.
We know that the trends are at least statistically significant at the $2\sigma$ uncertainty level, but the actual level is unknown to us. As



a conservative estimate we assume that the uncertainty is exactly at the $2\sigma$ level, consequently overestimating the actual trend
uncertainties. In red (blue) the trends derived from the FPH observations at Boulder are given for the time period from 1986
to 2010 (1988 to 2010). These were obtained by means of multi-linear regressions (see Eq. 2 later). Only small differences are
observed between two time periods. The trend estimates do not change significantly if the vertical resolution of the FPH data is
adjusted to that of the satellite observations. Likewise smoothing the FPH observations in time (with a 1 year running average),
to reduce the scatter among individual observations, does not notably affect the trend estimates.
The lower panel of Fig. 1 shows an estimate of the differences in the trend estimates between the FPH observations and
the merged zonal mean satellite data set. The differences vary with altitude ranging from about $0.25\,\mathrm{ppmv}\cdot\mathrm{decade}^{-1}$ to
$0.45\,\mathrm{ppmv}\cdot\mathrm{decade}^{-1}$. As argued above, the absolute size is probably underestimated while uncertainties are overestimated.
Given the importance of water vapour in the lower stratosphere there is a dire need to reconcile these differences. Potential
explanations could be the following or a combination of these:
(1)  There might be problems with one data set or even with both.
(2)  The location of Boulder might be not representative for the zonal mean due to, e.g. local processes specific for the location
(American monsoon, lee side of the Rocky Mountains etc.)
(3)  There might be unresolved differences between the measurement techniques, like due to the different spatial and temporal
sampling and resolution.
In their discussion of the trend discrepancies between the FPH observations and the merged satellite data set Hegglin et al.
(2014) opted for the second possible explanation, indicating that the temporal behaviour at Boulder is different than for the
zonal mean of the latitude band around the Boulder latitude. Trends derived from the CMAM simulations at $100\,\mathrm{hPa}$ (con-
sidering the time period 1980 to 2010) indicated longitudinal differences at $40°\mathrm{N}$, but also at other latitudes. Subsampling
the simulations to Boulder yielded better correlations with the FPH observations, in particular with respect to interannual
variations. Yet, the trends derived from the FPH observations and model simulations still disagreed, even in sign.
In this study we compare trend estimates for the Boulder location and the zonal mean of the latitude band around the Boulder
latitude using several model simulations and observational data sets. This aims to understand how much the sampling bias
contributes to the trend discrepancies shown in the lower panel of Fig. 1. The observations are meant to study this aspect on a
decadal scale while the simulations will be used to analyse even longer time periods. In the next section the model simulations
and observational data sets are briefly described. Section 3 outlines the analysis approach. The results of our analysis are
presented in Sect. 4 and subsequently discussed in Sect. 5.
**2   Data sets**
In our analysis we primarily utilise model simulations. We consider results from EMAC, WACCM, CMAM and CLaMS. On
the observational side we consider data from UARS/HALOE, Envisat/MIPAS and Aura/MLS. These data sets are analysed





individually to avoid potential uncertainties and artefacts due to merging (e.g. Ball et al., 2017), providing results for the time
periods 1992 – 2005, 2002 – 2012 and 2004 – 2016, respectively.

## 2.1 Model simulations

### 2.1.1 EMAC

The EMAC model is a numerical chemistry and climate simulation system that includes sub-models describing tropospheric
and middle atmosphere processes and their interaction with oceans, land and human influences (Jöckel et al., 2010). It uses the
second version of the Modular Earth Submodel System (MESSy2) to link multi-institutional computer codes. The core atmo-
spheric model is the $5^{th}$ generation European Centre Hamburg general circulation model (ECHAM5, Roeckner et al., 2006).
For the present study we applied EMAC (ECHAM5 version 5.3.02, MESSy version 2.50.5) in the T42L90MA-resolution,
i.e. with a spherical truncation of T42 (corresponding to a quadratic Gaussian grid of approximately 2.8° by 2.8° in latitude
and longitude) with 90 vertical hybrid pressure levels up to 0.01 hPa. The simulation was set up in accordance to the REF-
C1SD (transient hindcast reference simulation with specified dynamics) scenario defined in the framework of the SPARC
(Stratosphere-troposphere Processes And their Role in Climate) Chemistry-Climate Model Initiative (Eyring et al., 2013). Cor-
respondingly, it considers nudging (by Newtonian relaxation) towards data from the Interim ECMWF (European Centre for
Medium-Range Weather Forecasts) Reanalysis project (ERA-interim, Dee et al., 2011). Nudged parameters were the vorticity,
divergence, the logarithm of the surface pressure, the temperature and the mean temperature (wave number zero in spectral
space, Jöckel et al., 2016). Correspondingly, water vapour itself was not nudged and allowed to evolve freely. Depending on
parameter the nudging time constant varied between 6 h and 48 h. The initial conditions (in 1979) were taken from a corre-
sponding free-running simulation. In our analysis we use 10 hourly data, lasting until 2013.

### 2.1.2 WACCM

WACCM is an atmospheric component of the Community Earth System Model (CESM, Hurrell et al., 2013), a global climate
model with interactive ocean, sea ice, land and atmosphere. WACCM itself extends from the Earth's surface into the thermo-
sphere up to $5.1 \cdot 10^{-6}$ hPa (about 140 km). The simulation used 88 vertical levels and its horizontal resolution amounts to
1.9° in latitude and 2.5° in longitude (Marsh et al., 2013). As EMAC, the WACCM simulation employed here was set up ac-
cording to the REF-C1SD scenario. Meteorological fields from the MERRA (Modern Era Retrospective-Analysis for Research
and Applications, Rienecker et al., 2011) reanalysis data set were nudged from the surface to 50 km. Above 60 km the model
meteorological fields were fully interactive, with a linear transition in between. Here, temperature, zonal and meridional winds
and surface pressure were used to drive the physical parameterisation that control boundary layer exchanges, advective and

1    convective transport and the hydrological cycle. The nudging time constant used in this study was 50 h. The initial conditions

2    for the year 1979 were taken from a time-dependent REF-C1 simulation that started in 1955. Here we consider daily averaged

3    data and 2014 is the last year of the simulation.



### 2.1.3 CMAM

The Canadian Middle Atmosphere Model is a well-established and comprehensive chemistry climate model (de Grandpré et al., 2000; Scinocca et al., 2008). The CMAM simulation we employ is the same that has been used for the merging of the satellite data sets (Hegglin et al., 2014). It covers the period from 1979 to 2010 and provides results from the Earth's surface up to $0.0007\,hPa$ on 63 pressure levels. The horizontal resolution is $3.75°$ in latitude and longitude (T47). Horizontal winds and temperature data from ERA-interim were nudged up $1\,hPa$ with a nudging time constant of 24 hours at all levels. The nudging was performed in spectral space and only spectral coefficients up to T21 were nudged (McLandress et al., 2013, 2014). For the initial conditions the same simulation setup was run up to 1979, but nudging ERA-40 reanalysis data (Uppala et al., 2005). In our analysis we employ 6 hourly data.

### 2.1.4 CLaMS

The CLaMS model is fundamentally different to the models presented so far, as it is a Lagrangian chemistry transport model (McKenna et al., 2002b, a). It is driven by horizontal winds, temperature and diabatic heating rates that are taken from a reanalysis data set. CLaMS uses a hybrid vertical coordinate system which considers isentropes above about $300\,hPa$. The calculation of water vapour volume mixing ratios is based on simplified dehydration scheme (Ploeger et al., 2013). Below about $500\,hPa$ data from the driving reanalysis are used. Above, if saturation occurs along a trajectory the amount of water vapour in excess of the saturation ratio is frozen out and and partly sediments out, based on the fall speed of spherical ice particles of a mean size. Methane oxidation in the stratosphere is implemented using methane fields from the simulation and hydroxyl, oxygen and chlorine radicals from a model climatology. The simulation used in this work was driven by ERA-interim data. The results were interpolated on a regular pressure grid and use a horizontal resolution of $1°$ in latitude and longitude. We consider daily data (at 12 UTC) until 2010.

### 2.2 Observations

### 2.2.1 UARS/HALOE

HALOE was a solar occultation instrument deployed on UARS which was launched on 12 September 1991. Observations lasted until November 2005 shortly before the satellite was decommissioned. Based on the observation geometry 30 observations were performed per day. Those typically covered two distinct latitudes, one in the Northern and one in the Southern Hemisphere. Overall, latitudes between $80°S$ and $80°N$ were covered. HALOE measured in the infrared spectral region covering some specific bands between $2.5\,\mu m$ and $11\,\mu m$. Water vapour information has been retrieved from a spectral band ranging from $6.54\,\mu m$ to $6.67\,\mu m$, typically covering altitudes from the upper troposphere to the upper mesosphere. In this study we employ data derived with retrieval version 19 (Kley et al., 2000). Occultations with anomalies regarding the trip angle (http://haloe.gats-inc.com/user_docs/events_terminate_below_150km.pdf) and the lockdown angle (http://haloe.gats-inc.com/user_docs/smoothed_





lockdown_angles.pdf) were screened out. Also observations before March 1992 were discarded as they might be affected by
aerosols from the Pinatubo volcanic eruption in June 1991.

### 2.2.2   Envisat/MIPAS

MIPAS was a high-resolution Fourier transform spectrometer flown on Envisat. The satellite was launched on 1 March 2002 and
operated until 8 April 2012. The MIPAS instrument measured thermal emission in the infrared spectral region between $4.1\,\mu m$
and $14.6\,\mu m$ covering the entire latitude range (Fischer et al., 2008). Initially the measurements used a spectral resolution of
$0.025\,cm^{-1}$ (unapodised). Due to an instrument failure in March 2004 the spectral resolution had to be reduced to $0.0625\,cm^{-1}$.
Observations with the lower spectral resolution recommenced in January 2005. In accordance the MIPAS time period is split in
two periods which are referred to as full (FR) and reduced (RR) resolution period. During the FR period more than 1000 scans
were performed daily while during the RR period it were more than 1300 scans. Water vapour information is retrieved from
12 microwindows between $6.3\,\mu m$ and $12.6\,\mu m$ typically covering the upper troposphere to the middle mesosphere. Here we
combine data from the retrieval version 20 for the FR period and version 220/221 for the RR period (Schieferdecker et al.,
2015; Lossow et al., 2017), both generated with the research processor operated at IMK/IAA (Institut für Meteorologie und
Klimaforschung (IMK) in Karlsruhe, Germany / Instituto de Astrofísica de Andalucía (IAA) in Granada, Spain). The overall
time period ranges from July 2002 to April 2012. Before the analysis the data were screened considering the visibility flag
and averaging kernel diagonal criterion (discard data with diagonal values $< 0.03$). The former flags data below the lowermost
usable tangent altitude while the latter criterion concerns the measurement contribution to the retrieved data.

### 2.2.3   Aura/MLS

The Microwave Limb Sounder is an instrument aboard NASA's (National Aeronautics and Space Administration) Aura satel-
lite. The satellite was launched on 15 July 2004 and uses a sun-synchronous orbit, as Envisat did. The MLS instrument measures
microwave thermal emission at the limb of the Earth's atmosphere, covering the latitude range between 82°S and 82°N. An
atmospheric scan takes about $25\,s$, resulting in more than 3400 observations per day (Waters et al., 2006). Water vapour infor-
mation is derived from the strong emission line centred at $183\,GHz$, covering the altitude range from the upper troposphere
to the upper mesosphere. In the analysis we used data from the latest retrieval version 4.2, considering the time period from
August 2004 to December 2016. Prior any analysis the data were screened according to the criteria listed in the data quality
document (Livesey et al., 2015).

### 2.2.4   NOAA frost point hygrometer

For the sake of completeness we provide here also a more detailed description of the NOAA frost point hygrometer. The FPH
measurement principle is based on maintaining a thin, stable layer of frost on a chilled mirror as air flows past it at $5\,m \cdot s^{-1}$.
Stability in frost coverage is detected optically and maintained by rapidly adjusting the mirror temperature. When the frost
coverage is stable, the ice and overlying water vapour are in equilibrium and the ice surface temperature (frost point tempera-



ture) is directly related to the partial pressure of water vapour in the air stream. At 50 hPa, a 0.5 ppmv (about 10%) change in
the water vapour mixing ratio produces a 0.42 K change in the frost point temperature. The mirror temperature is measured by
a thermistor calibrated to an accuracy better than 0.05 K. Hall et al. (2016) provide detailed descriptions of the instrument and
its history, along with an assessment of its measurement uncertainties. The primary measurement uncertainty is related to in-
stabilities in frost coverage that can produce frost point temperature errors as large as $\pm0.5$ K in the stratosphere. However, the
instabilities are generally oscillatory in nature and therefore manifest as random errors, not systematic biases. Each thermistor
is meticulously calibrated against a temperature probe certified by the National Institute of Standards and Technology (NIST)
and, to ensure calibration stability over the long term (i.e. decades), a small archive of previously calibrated thermistors. Total
FPH measurement uncertainties (95% confidence) in the stratosphere are estimated to be smaller than 0.3 ppmv (about 6%,
Hall et al., 2016). The 30 year net increase ($\sim$1 ppmv, see Introduction) in stratospheric water vapour observed over Boulder
translates to a 0.8 K rise in frost point temperatures that greatly exceeds the FPH measurement uncertainties.

## 3 Approach

### 3.1 Boulder time series

For the Boulder time series we consider simulated data and satellite observations that are spatially located within:
• a 1000 km radius around the Boulder FPH observation site.
• the latitude band between 35°N – 45°N.
In the analysis of the HALOE data set we use less strict criteria because of its sparseness relative to the other data sets.
Instead of the radius criterion data in the wider longitude range between 130°W and 80°W are considered.
In temporal terms we consider for the Boulder time series two sets of data. Set #1 simply comprises all data in a given month.
We will refer to these time series as full time series. Set #2 is adapted to the individual FPH observations at Boulder. From that
we can also assess the role of the temporal component of the sampling bias. For the simulations the data from the closest time
step are used. For the observations all data obtained within $\pm12$ h of the FPH measurements are considered. These time series
we will refer to as adapted time series.
All data obeying the spatial and temporal criteria are combined to monthly means. For the observations we consider only
monthly means that are based on at least 5 measurements to avoid spurious results. As a result, a temporal adaption to the
individual FPH observations is only meaningful for the MLS observations.

### 3.2 Zonal mean time series

For the zonal mean time series we consider monthly means of all data in the latitude range between 35°N to 45°N, resembling
the merged satellite data set. Monthly zonal means derived from the satellite observations are discarded if they are not based
on a minimum number of 20 measurements. If a monthly mean does not exist for the Boulder time series, e.g. because there



were no FPH observations for the adapted time series or due to screening of the satellite data, this monthly mean is also not
considered for the zonal mean results.
In addition, we also investigate how the trend estimates at Boulder compare to those for zonal means of other latitude bands.
For that we consider the latitude bands 45°N – 55°N, 25°N – 35°N, Equator – 60°N and 60°S – 60°N. The first two bands
are adjacent to the latitude band around the Boulder latitude. The last two bands cover a wider range of latitudes. This aims to
investigate how representative trends at Boulder are on regional and more global scales.
**3.3  De-seasonalisation**
In our analysis we employ de-seasonalised data. This enhances the visibility of the long-term behaviour and has the positive
side effect that the MIPAS observations from the FR and RR periods are homogenised. Typically between these periods a small
bias in the absolute water vapour volume mixing ratios exists. The de-seasonalisation is achieved by means of regression, again
motivated by the MIPAS data. This approach has the advantage of working for time series that cover a time period between
12 months and 24 months, which applies here to the MIPAS data for the FR period. The regression model contains an offset
and a parametrisation for the semi-annual (SAO) and annual variation (AO) using orthogonal sine and cosine functions:

$$
\begin{aligned}
f_{\mathrm{d}}(t,\phi,z) = {} & C_{\mathrm{offset}}(\phi,z) + \\
& C_{\mathrm{SAO_1}}(\phi,z) \cdot \sin(2 \cdot \pi \cdot t / p_{\mathrm{SAO}}) + \\
& C_{\mathrm{SAO_2}}(\phi,z) \cdot \cos(2 \cdot \pi \cdot t / p_{\mathrm{SAO}}) + \\
& C_{\mathrm{AO_1}}(\phi,z) \cdot \sin(2 \cdot \pi \cdot t / p_{\mathrm{AO}}) + \\
& C_{\mathrm{AO_2}}(\phi,z) \cdot \cos(2 \cdot \pi \cdot t / p_{\mathrm{AO}}).
\end{aligned}
\tag{1}
$$

In the equation $f_{\mathrm{d}}(t,\phi,z)$ denotes the fit of the regressed time series for a given time $t$ (in years), latitude band $\phi$ and al-
titude $z$ which is subsequently subtracted from the absolute time series to obtain the de-seasonalised time series. $C$ are the
regression coefficients of the individual model components and $p_{\mathrm{SAO}}$ and $p_{\mathrm{AO}}$ represent the time periods of the semi-annual
(0.5 years) and annual variation (1 year), respectively. The regression coefficients are derived according to the method outlined
by von Clarmann et al. (2010), using the standard errors of the monthly means (their inverse squared) as statistical weights.
Autocorrelation effects and empirical errors (Stiller et al., 2012) are not considered in this regression.
For the de-seasonalisation of the simulations we consider data in the time period from 1985 to 2010. The start year is
chosen because of obvious differences in the water vapour abundances among the simulations, related to differences in their
initial conditions and spin up time (see Fig. 2 and Sect. 4.1). 2010 is the last year that is covered by all simulations. For the
observations it is not possible to use a consistent time period. Instead the entire time period covered by the individual data sets
is used for the de-seasonalisation.





## 3.4 Trend estimates and trend differences

Like the de-seasonalisation of the time series, the estimation of the water vapour trends is based on regression. For this analysis the regression model is as follows:

$$
\begin{aligned}
f_t(t, \phi, z) = & C_{\text{offset}}(\phi, z) + C_{\text{trend}}(\phi, z) \cdot t + \\
& C_{\text{SAO}_1}(\phi, z) \cdot \sin(2 \cdot \pi \cdot t / p_{\text{SAO}}) + \\
& C_{\text{SAO}_2}(\phi, z) \cdot \cos(2 \cdot \pi \cdot t / p_{\text{SAO}}) + \\
& C_{\text{AO}_1}(\phi, z) \cdot \sin(2 \cdot \pi \cdot t / p_{\text{AO}}) + \\
& C_{\text{AO}_2}(\phi, z) \cdot \cos(2 \cdot \pi \cdot t / p_{\text{AO}}) + \\
& C_{\text{QBO}_1}(\phi, z) \cdot QBO_1(t) + \\
& C_{\text{QBO}_2}(\phi, z) \cdot QBO_2(t).
\end{aligned}
\tag{2}
$$

In comparison to the regression model used for the de-seasonalisation, it contains in addition a trend term $C_{\text{trend}}$ and a parametrisation for the quasi-biennial oscillation (QBO). In our analysis we determine only a single trend for the entire time period. Trend changes within this period are correspondingly not analysed (see e.g. Hurst et al., 2011). Even though the regression is applied to de-seasonalised time series the SAO and AO terms are kept since the regression models for the de-seasonalisation and trend analysis differ. The QBO parametrisation is based on normalised winds at $50\,\text{hPa}$ ($QBO_1$) and $30\,\text{hPa}$ ($QBO_2$) observed over Singapore (1°N, 104°E), which are almost orthogonal. These data are provided by Freie Universität Berlin (webpage: http://www.geo.fu-berlin.de/met/ag/strat/produkte/qbo/qbo.dat). Unlike for the de-seasonalisation, in this regression we consider autocorrelation effects and empirical errors (Stiller et al., 2012), to obtain optimal estimates of the trends and their uncertainties.

To be consistent with our motivation shown in Fig. 1, we calculate the water vapour trends separately for the Boulder time series and the zonal mean time series and subsequently derive the trend differences. Correspondingly, the trend differences ($\Delta C_{\text{trend}}$) and their uncertainties ($\varepsilon_{\text{trend}}$) are given as:

$$
\begin{aligned}
\Delta C_{\text{trend}}(\phi, z) &= C_{\text{trend}}^{\text{Boulder}}(z) - C_{\text{trend}}^{\text{zonal mean}}(\phi, z) \\
\varepsilon_{\text{trend}}(\phi, z) &= \sqrt{\varepsilon_{\text{trend}}^{\text{Boulder}}(z)^2 + \varepsilon_{\text{trend}}^{\text{zonal mean}}(\phi, z)^2}
\end{aligned}
\tag{3}
$$

Here $C_{\text{trend}}^{\text{Boulder}}$ represents the trends derived from the Boulder time series and $\varepsilon_{\text{trend}}^{\text{Boulder}}$ are the corresponding uncertainties. Likewise $C_{\text{trend}}^{\text{zonal mean}}$ and $\varepsilon_{\text{trend}}^{\text{zonal mean}}$ denote the trends calculated from the zonal mean time series and their uncertainties.





## 4 Results

In this section we will first present the simulation results and subsequently the results derived from the observations. We focus on the altitude range between $100\,\mathrm{hPa}$ and $20\,\mathrm{hPa}$ that is typically covered by the FPH observations and in almost all cases completely entirely in the stratosphere (Kunz et al., 2013).

### 4.1 Simulations

Figure 2 shows for the different model simulations the de-seasonalised Boulder time series (top panel) and the zonal mean time series around the Boulder latitude (latitude range $35°\mathrm{N} - 45°\mathrm{N}$, middle panel) at $70\,\mathrm{hPa}$. The differences between the two time series are shown in the lower panel as a complement. The time series adapted to the individual FPH observations (see Sect. 3.1 and 3.2) are marked with the suffix (A) in the figure legend. Overall, the Boulder and the zonal mean time series are visually rather similar, with the latter being more smooth. The difference time series show occasionally larger deviations (up to $0.6\,\mathrm{ppmv}$ in absolute terms), however any conspicuous behaviour or a trend appear to be absent. In general, the different simulations yield similar results for Boulder and the zonal mean. The most prominent exception is observed in the early 1980s. This relates to differences in the 1979 initial conditions and the spin up times among the simulations. Until 1985 the EMAC anomalies are significantly lower than for the other simulations. In the first years the largest anomalies are found in the CMAM results, which were probably caused by higher water vapour volume mixing ratios in the initial conditions based on the nudging of ERA-40 data (see Sect. 2.1.3). Presumably the best representation is provided by CLaMS which, as a Lagrangian model, does not need to deal with these aspects.

Figure 3 shows the trend estimates for the time series at Boulder (left column) and the zonal mean for the latitude band between $35°\mathrm{N}$ and $45°\mathrm{N}$ (middle column). The right column shows the corresponding difference according to Eq. 3. The different rows consider different time periods, i.e. $1985 - 2010$ (top row), $1990 - 2010$ (middle row) and $1995 - 2010$ (bottom row), as also indicated in the title of the centre panels. We have not included the time period from 1980 to 2010 here. The differences in the water vapour anomalies among the simulations in the early 1980s primarily affect the trends for Boulder and the zonal mean, yet the trend differences are comparable to those for the time period from 1985 to 2010. Trends and trend differences significant at the $2\sigma$ uncertainty levels are marked by triangles.

For the time period between 1985 and 2010, the EMAC results exhibit positive trend estimates at Boulder. They range between $0.04\,\mathrm{ppmv}\cdot\mathrm{decade}^{-1}$ and $0.12\,\mathrm{ppmv}\cdot\mathrm{decade}^{-1}$. The trends derived from the adapted time series yield smaller values as those obtained from the full time series, by about $0.02\,\mathrm{ppmv}\cdot\mathrm{decade}^{-1}$ to $0.03\,\mathrm{ppmv}\cdot\mathrm{decade}^{-1}$. The results derived from the other simulations indicate rather small trends at Boulder, typically within $\pm0.05\,\mathrm{ppmv}\cdot\mathrm{decade}^{-1}$. A few differences exist in the actual sign of the trends, most prominently above $40\,\mathrm{hPa}$ where the WACCM and CMAM results indicate negative trends (around $-0.04\,\mathrm{ppmv}\cdot\mathrm{decade}^{-1}$) while they are positive for CLaMS (up to around $0.05\,\mathrm{ppmv}\cdot\mathrm{decade}^{-1}$ at $20\,\mathrm{hPa}$). The results derived from the full and the adapted time series exhibit small quantitative differences. Typically they are of a similar size as those described above for the EMAC results. Overall, the spread among the trend estimates ranges from about $0.10\,\mathrm{ppmv}\cdot\mathrm{decade}^{-1}$ at $100\,\mathrm{hPa}$ to $0.16\,\mathrm{ppmv}\cdot\mathrm{decade}^{-1}$ at $20\,\mathrm{hPa}$. The trend estimates derived from the zonal mean time



series for the latitude band between 35°N and 45°N look very similar to those derived for Boulder. Correspondingly, the
trend differences between Boulder and the zonal mean are very small. The differences never exceed $0.04\,\mathrm{ppmv \cdot decade^{-1}}$ in
absolute terms. The largest differences are derived for EMAC and WACCM (based on the adapted time series) at $100\,\mathrm{hPa}$. The
trend differences are predominantly positive below $70\,\mathrm{hPa}$ and mostly negative above $50\,\mathrm{hPa}$. The exact altitude dependence
differs in details among the different simulation results.
Both for Boulder and the zonal mean, the trend estimates for the time period from 1990 to 2010 are negative. There
are differences among the individual simulations. The agreement is, however, better than for the time period between 1985
and 2010. The spread maximises at $100\,\mathrm{hPa}$ with about $0.12\,\mathrm{ppmv \cdot decade^{-1}}$ and is smallest above $40\,\mathrm{hPa}$ with about
$0.06\,\mathrm{ppmv \cdot decade^{-1}}$. Differences among the results derived from the full and adapted time series are very small. The trend
differences between Boulder and the zonal mean are of similar size as for the time period 1985 to 2010. A similar behaviour
in terms of the altitude dependence is also visible.
The last time period we consider is from 1995 to 2010. At $100\,\mathrm{hPa}$ consistently positive trends are found, except for the
adapted EMAC time series. Overall, the trend estimates vary between $-0.02\,\mathrm{ppmv \cdot decade^{-1}}$ and $0.14\,\mathrm{ppmv \cdot decade^{-1}}$.
With increasing altitude the trend estimates typically decrease and above $45\,\mathrm{hPa}$ they are all negative. The decrease continues
up to $20\,\mathrm{hPa}$, except for the CMAM results which indicate a slight increase above $40\,\mathrm{hPa}$. At $20\,\mathrm{hPa}$ the trend estimates vary
between $-0.24\,\mathrm{ppmv \cdot decade^{-1}}$ and $-0.08\,\mathrm{ppmv \cdot decade^{-1}}$ among the simulations, with significantly smaller differences
between the results derived from the full and the adapted time series. The best agreement among the simulations is observed
around $80\,\mathrm{hPa}$ where the spread is about $0.08\,\mathrm{ppmv \cdot decade^{-1}}$. The altitude dependence and the spread among the simulations
is similar for the trend estimates derived from the zonal mean time series. Quantitatively there are larger differences between
the Boulder and zonal mean trends, clearly surpassing those observed for the other time periods. Above $60\,\mathrm{hPa}$ the differences
are still within $\pm 0.02\,\mathrm{ppmv \cdot decade^{-1}}$. Below this altitude the differences occasionally exceed $\pm 0.05\,\mathrm{ppmv \cdot decade^{-1}}$. The
largest trend differences are derived from the adapted EMAC and CMAM time series.
To expand on the temporal development of the trend differences between Boulder and the zonal mean even more we derive
these differences continuously for 11 year periods, as shown in Fig. 4. This aims also to build a bridge to the shorter observa-
tional results that will be presented in Sect. 4.2. The results are assigned to the centre of the considered period, e.g. to 1995
for the time period between 1990 and 2000. The trend differences vary with time and altitude in size and sign. On this shorter
time scale the differences are typically larger than observed for the longer time periods described in the last figure. There is
also a more prominent distinction between the results derived from the full and the adapted time series. The latter yield larger
differences on an absolute scale, but also some patterns are different.

6       For the full time series the trend differences are generally within $\pm 0.04\,\mathrm{ppmv \cdot decade^{-1}}$. Exceptions from this behaviour are

primarily observed at the lowermost altitudes. In particular the EMAC results exhibit significantly larger differences, increasing
to about $\pm 0.15\,\mathrm{ppmv \cdot decade^{-1}}$ at $100\,\mathrm{hPa}$. The temporal development of the trend differences exhibits a number of common
features among the simulations, even though quantitative differences are obvious. At the lowermost altitudes all simulations
show negative trend differences from 1990 to about 1999. Afterwards positive trend differences are found. Higher up, i.e. above
about $50\,\mathrm{hPa}$, positive trend differences are visible from 1995 to about 2004.





The trend differences derived from the adapted time series are within $\pm 0.08\,\mathrm{ppmv \cdot decade^{-1}}$ above $60\,\mathrm{hPa}$. Below, they
increase again to about $0.2\,\mathrm{ppmv \cdot decade^{-1}}$ in absolute terms. The different simulations agree on some difference patterns, as
observed for the results derived from the full time series. Most prominently, above $50\,\mathrm{hPa}$ the trend differences are typically
positive from about 1990 to 2003 and negative afterwards. The bisection of trend differences at the lowermost altitudes derived
from the full time series is only visible in some simulations. Finally, it should be noted, that none of the trend differences shown
in Fig. 4 are statistically significant at the $2\sigma$ uncertainty level.
To investigate the representativeness of the Boulder trends on a larger geographical scale Fig. 5 compares them to the trends
derived from zonal mean time series from five latitude bands, namely $35°N - 45°N$ (row #1), $45°N - 55°N$ (row #2), $25°N -$
$35°N$ (row #3), Equator $- 60°N$ (row #4) and $60°S - 60°N$ (row #5). The figure considers the time period between 1987
and 2010, approximately the time coverage of the merged zonal mean satellite data set. The results in the left column are the
same for all rows and kept for the sake of convenience. The trends at Boulder are close to those obtained for the time periods
1985 – 2010 and 1990 – 2010 shown in Fig. 3. Note, that in Fig. 5 the x-axis is smaller, allowing a more detailed picture.
Overall, the trends at Boulder are within $\pm 0.07\,\mathrm{ppmv \cdot decade^{-1}}$. Clear differences among the simulations exist, while the
differences between the results derived from the full and the adapted time series are typically smaller. The EMAC results
indicate positive trends (up to almost $0.1\,\mathrm{ppmv \cdot decade^{-1}}$). Statistical significance is only visible at the highest altitudes.
The trend estimates derived from the adapted time series are again smaller than those determined from the full time series.
With few exceptions the same behaviour is also observed for all other simulations. The trends derived from CLaMS are
negative below $35\,\mathrm{hPa}$ and positive above, ranging from about $-0.05\,\mathrm{ppmv \cdot decade^{-1}}$ to $0.05\,\mathrm{ppmv \cdot decade^{-1}}$. They are
relatively constant up to $60\,\mathrm{hPa}$ before they start to increase significantly. The WACCM and CMAM trends show a similar
altitude dependence with maximum negative trends (around $-0.05\,\mathrm{ppmv \cdot decade^{-1}}$) in the altitude range between $50\,\mathrm{hPa}$
and $40\,\mathrm{hPa}$. For WACCM the trend estimates become positive below $80\,\mathrm{hPa}$ while those derived from the CMAM simulations
are negative at all altitudes.
As observed in Fig. 3 the trends derived from the zonal mean time series for the latitude band between $35°N$ and $45°N$ are
very similar to those for Boulder. Correspondingly, the trend differences are small, i.e. ranging from $-0.02\,\mathrm{ppmv \cdot decade^{-1}}$
to $0.04\,\mathrm{ppmv \cdot decade^{-1}}$. The differences are typically positive below $70\,\mathrm{hPa}$ and mostly negative above, affirming the picture
observed for the time periods 1985 – 2010 and 1990 – 2010 in Fig. 3.
The trends derived from the zonal mean time series for the other latitude bands exhibit many common features with the
results for the zonal mean between $35°N$ and $45°N$. There are quantitative changes, but overall the trend estimates remain of
the same order. Besides that, also the altitude dependence of the trends remains very similar and so do the relations among
the different simulations. The trend differences between Boulder and the zonal mean for $45°N$ and $55°N$ remain within
$\pm 0.04\,\mathrm{ppmv \cdot decade^{-1}}$. Again, the differences are typically positive below $70\,\mathrm{hPa}$ and predominantly negative at higher
altitudes. The trend differences between Boulder and the zonal means for $25°N$ to $35°N$ and from the Equator to $60°N$ are
quite similar, at least up to about $35\,\mathrm{hPa}$. In both cases the trend differences are within $\pm 0.03\,\mathrm{ppmv \cdot decade^{-1}}$. Typically
the EMAC and CLaMS results are at the higher end of this interval while the WACCM and CMAM results are at the lower
end. The largest trend differences compared to Boulder are observed for the zonal mean of the latitude band between $60°S$





and 60°N. These range from $-0.04\,\mathrm{ppmv}\cdot\mathrm{decade}^{-1}$ to slightly more than $0.06\,\mathrm{ppmv}\cdot\mathrm{decade}^{-1}$. There is clear separation
between the CLaMS results and those from the other simulations. For the CLaMS simulation the trend differences are negative
at $100\,\mathrm{hPa}$ (around $-0.015\,\mathrm{ppmv}\cdot\mathrm{decade}^{-1}$). Around $90\,\mathrm{hPa}$ they turn positive and continue to increase within increasing
altitude. At $20\,\mathrm{hPa}$ the differences amount to $0.05\,\mathrm{ppmv}\cdot\mathrm{decade}^{-1}$ for the adapted time series and $0.06\,\mathrm{ppmv}\cdot\mathrm{decade}^{-1}$
for the full time series, respectively. The other simulations indicate positive trend differences at $100\,\mathrm{hPa}$. Around $70\,\mathrm{hPa}$ the
differences become negative and peak in absolute size between $50\,\mathrm{hPa}$ and $40\,\mathrm{hPa}$ (between $-0.04\,\mathrm{ppmv}\cdot\mathrm{decade}^{-1}$ and
$-0.02\,\mathrm{ppmv}\cdot\mathrm{decade}^{-1}$). Higher up, the trend differences get less negative again.
**4.2   Observations**
Figure 6 shows for the HALOE, MIPAS and MLS observations the de-seasonalised Boulder time series (top panel) and the
zonal mean time series around the Boulder latitude at $70\,\mathrm{hPa}$ (middle panel). As in Fig. 2 the lower panel shows the differences
between the two time series in addition. For MLS there is also a data set that is adapted to the FPH observations at Boulder (see
Sect. 3.1). Like the simulations the observations exhibit a rather similar picture for Boulder and the zonal mean. The difference
time series indicate occasionally some larger deviations. For example in the second half of 2011 some substantial positive
differences are observed, consistent in the MIPAS and MLS data. The largest differences typically occur in the MLS data set
that is adapted to the FPH observations and for HALOE data set, primarily due to its sparseness. In addition, there is a notable
agreement between the MIPAS and MLS time series for Boulder and the zonal mean time series.
Figure 7 compares the trend estimates at Boulder with those derived from zonal mean time series for various latitude bands.
The results for the different observational data sets consider different time periods as indicated in the figure legend. Thus, they
are not comparable and will be addressed separately.
In the lower stratosphere the HALOE observations exhibit negative trends at Boulder for the time period between 1992
and 2005. This behaviour is primarily related to the significant drop in lower stratospheric water vapour in 2001 (Randel
et al., 2006; Scherer et al., 2008; Brinkop et al., 2016). The relative dryness continued until 2005 (coinciding with the end of
the HALOE observations), causing the 13 year HALOE trends to be negative. The largest trends are observed below $80\,\mathrm{hPa}$
with values around $-0.45\,\mathrm{ppmv}\cdot\mathrm{decade}^{-1}$. Above, the trends get less negative with increasing altitude. At $20\,\mathrm{hPa}$ the trend
amounts to about $-0.03\,\mathrm{ppmv}\cdot\mathrm{decade}^{-1}$ (and is not statistically significant). The trends derived for the zonal mean between
$35°\mathrm{N}$ to $45°\mathrm{N}$ have a similar altitude dependence, but their absolute sizes are smaller. Accordingly, the trend differences
between Boulder and the zonal mean are negative. Above $80\,\mathrm{hPa}$ the differences are almost invariant with altitude. Here they
amount to about $-0.05\,\mathrm{ppmv}\cdot\mathrm{decade}^{-1}$. At lower altitudes the differences are larger maximising at $100\,\mathrm{hPa}$ with about
$-0.1\,\mathrm{ppmv}\cdot\mathrm{decade}^{-1}$. For the other latitude bands the zonal mean trends exhibit the same kind of altitude dependence as
observed for the band from $35°\mathrm{N}$ to $45°\mathrm{N}$. The most prominent variation concerns the exact altitude in which the negative
trends exhibit their absolute maximum. For the latitude band between $45°\mathrm{N}$ and $55°\mathrm{N}$ this occurs close to $90\,\mathrm{hPa}$. For the
trends derived from the zonal mean from the Equator to $60°\mathrm{N}$ and from $60°\mathrm{S}$ to $60°\mathrm{N}$ this maximum is observed around
$70\,\mathrm{hPa}$. The trend differences between Boulder and the zonal mean for the latitude band between $45°\mathrm{N}$ and $55°\mathrm{N}$ range from
$-0.1\,\mathrm{ppmv}\cdot\mathrm{decade}^{-1}$ to $0\,\mathrm{ppmv}\cdot\mathrm{decade}^{-1}$ with the largest absolute values occurring below $75\,\mathrm{hPa}$. For the latitude band



between 25°N and 35°N the trend differences are close to zero, except below 75 hPa where they become significantly more negative. For the remaining two latitude bands the trend differences are quite similar. At 100 hPa the trend differences amount to $-0.15\,\mathrm{ppmv}\cdot\mathrm{decade}^{-1}$. Towards 60 hPa the differences increase to around $0.05\,\mathrm{ppmv}\cdot\mathrm{decade}^{-1}$. Between 60 hPa and 30 hPa the trend differences are rather constant. Higher up, they increase to more than $0.1\,\mathrm{ppmv}\cdot\mathrm{decade}^{-1}$.

The MIPAS observations indicate positive trends at Boulder during the time period from 2002 to 2012. The trends decrease with increasing altitude from about $0.25\,\mathrm{ppmv}\cdot\mathrm{decade}^{-1}$ at 100 hPa to $0.1\,\mathrm{ppmv}\cdot\mathrm{decade}^{-1}$ at 20 hPa. For the zonal mean between 35°N and 45°N the trend estimates are also consistently positive. However, they show a slightly different altitude dependence than for Boulder. Below about 70 hPa the trends increase while higher up they decrease. In correspondence, the trend differences between the Boulder and zonal mean estimates are most pronounced below about 70 hPa, rising to $0.05\,\mathrm{ppmv}\cdot\mathrm{decade}^{-1}$ at 100 hPa. Above 70 hPa the differences are close to zero. A very similar behaviour is observed for the trend differences between Boulder and the zonal mean considering the latitude band between 45°N and 55°N. The trend differences to the estimates for the latitude bands from 25°N to 35°N and the Equator to 60°N exhibit a pronounced altitude dependence. They decrease from more than $0.1\,\mathrm{ppmv}\cdot\mathrm{decade}^{-1}$ at 100 hPa to $-0.05\,\mathrm{ppmv}\cdot\mathrm{decade}^{-1}$ at 20 hPa. The sign of the trend differences switches at about 60 hPa. The trend differences between Boulder and the zonal mean for 60°S to 60°N are positive at all altitudes. The smallest differences are close to zero and are observed between 45 hPa and 30 hPa. The largest difference is visible at 100 hPa with $0.15\,\mathrm{ppmv}\cdot\mathrm{decade}^{-1}$.

The Boulder trends derived from the MLS observations from 2004 to 2016 are positive. They exhibit a pronounced altitude dependence. The trend estimates exhibit maxima at 70 hPa ($0.4\,\mathrm{ppmv}\cdot\mathrm{decade}^{-1}$) and 30 hPa (close to $0.3\,\mathrm{ppmv}\cdot\mathrm{decade}^{-1}$). Minima are found at 100 hPa ($0.2\,\mathrm{ppmv}\cdot\mathrm{decade}^{-1}$), 45 hPa and 20 hPa (around $0.25\,\mathrm{ppmv}\cdot\mathrm{decade}^{-1}$). The trends derived from the adapted time series are slightly larger than those calculated from the full time series. The trend differences between these two data sets are of a similar order as observed for the simulations addressed before. The MLS trends derived from the zonal mean time series for the different latitudes indicate a similar altitude dependence as that observed for Boulder. Overall, the trend differences between Boulder and the zonal means are generally within $\pm 0.05\,\mathrm{ppmv}\cdot\mathrm{decade}^{-1}$. Prominent exceptions occur below 70 hPa for the differences to the zonal means from 25°N to 35°N, Equator to 60°N and 60°S to 60°N. Here, the differences can be as large as $0.15\,\mathrm{ppmv}\cdot\mathrm{decade}^{-1}$. In addition, the trend differences between Boulder and the zonal mean from 60°S to 60°N are noticeably larger than for the other latitude bands, ranging from $0.05\,\mathrm{ppmv}\cdot\mathrm{decade}^{-1}$ to $0.15\,\mathrm{ppmv}\cdot\mathrm{decade}^{-1}$. Beyond that, the trend differences are consistently larger (by about $0.05\,\mathrm{ppmv}\cdot\mathrm{decade}^{-1}$) for the adapted time series at altitudes around 40 hPa.

## 5   Discussion and conclusions

In this work we compared trend estimates for lower stratospheric water vapour between Boulder and zonal mean data around the Boulder latitude (35°N to 45°N), considering different simulations and observations. The primary objective was to verify if sampling biases could possibly help to explain the discrepancies in the trend estimates between the FPH observations at Boulder (Hurst et al., 2011) and a merged zonal mean satellite data set (Hegglin et al., 2014). For the time period from the late





1980s to 2010 the trend differences (FPH minus merged zonal mean satellite data set) range from $0.25\,\mathrm{ppmv}\cdot\mathrm{decade}^{-1}$ to
$0.45\,\mathrm{ppmv}\cdot\mathrm{decade}^{-1}$, increasing with altitude.
Our analysis shows that there are differences in the trend estimates between Boulder and the zonal mean, both for the
simulations and observations. These trend differences are dependent on altitude and the time period considered. For the
time period from the late 1980s to 2010 the simulations indicate trend differences between about $-0.02\,\mathrm{ppmv}\cdot\mathrm{decade}^{-1}$
to $0.04\,\mathrm{ppmv}\cdot\mathrm{decade}^{-1}$ (which are however not statistically significant different from zero). These are clearly smaller than
the discrepancies in the trend estimates derived from the FPH observations and the merged satellite data set. The larger positive
differences are observed close to $100\,\mathrm{hPa}$. Here, the sampling bias partly resolves the observational discrepancies. Above about
$60\,\mathrm{hPa}$ the trend differences derived from the model simulations are however typically negative. This indicates that the trend
estimates for the zonal mean data should be larger than at Boulder, which is contradictory to the observed trend differences
between the FPH observations and the merged zonal mean satellite data set. For the time period from the late 1980s to 2010,
the simulations also do not indicate any significant deviations in the trend differences derived from time series using all data
during a given month (which we referred to as full time series) or just using that closest in time to the actual FPH observations
(which we referred to as adapted time series). This indicates that the temporal contribution to the sampling bias on this time
scale is small.
Given these model results, a sampling bias is not a viable explanation for the trend discrepancies between the FPH obser-
vations at Boulder and the merged zonal mean satellite data set presented in Hegglin et al. (2014). It still could be the case
that the simulations underrepresent variability or that the trend differences originate from smaller spatial and temporal scales
than are resolved by the model simulations (i.e. sub-grid processes). For the Boulder time series we used data in a $1000\,\mathrm{km}$
radius around the Boulder FPH observation site and within the latitude range from $35°\mathrm{N}$ to $45°\mathrm{N}$. These criteria were primarily
chosen for consistency with the analysis of the satellite observations whose exact measurement locations vary from orbit to
orbit and day to day. In an additional analysis of the simulations, we considered for the Boulder time series only data from the
closest grid point in space (EMAC: $40.5°\mathrm{N}$, $104.1°\mathrm{W}$, $\Delta r=109\,\mathrm{km}$; WACCM: $40.7°\mathrm{N}$, $105.0°\mathrm{W}$, $\Delta r=80\,\mathrm{km}$; CMAM: $39.0°\mathrm{N}$,
$105.0°\mathrm{W}$, $\Delta r=113\,\mathrm{km}$; CLaMS: $40.0°\mathrm{N}$, $105.0°\mathrm{W}$, $\Delta r=17\,\mathrm{km}$). This analysis yields small quantitative changes (not shown).
Qualitatively, exactly the same conclusions can be drawn as from the standard analysis. The temporal resolutions of the anal-
ysed simulations vary (see Sect. 2.1). The CMAM simulations provide the best resolution in this analysis with $6\,\mathrm{h}$. Accordingly
the worst temporal mismatch to the actual FPH observations is $3\,\mathrm{h}$. This gives an upper limit of temporal scales not covered
in this analysis. Yet, arguably the different simulations yield similar results, as do the analysis of the full and the adapted time
series.
For a single decade of data the trend differences between Boulder and the $35°\mathrm{N}$ – $45°\mathrm{N}$ zonal mean are typically larger
than those discussed above for the entire time period from the late 1980s to 2010. The differences are typically within
$\pm 0.10\,\mathrm{ppmv}\cdot\mathrm{decade}^{-1}$, except close to $100\,\mathrm{hPa}$ where the differences can be occasionally as large as $\pm 0.2\,\mathrm{ppmv}\cdot\mathrm{decade}^{-1}$.
For the simulations, the trend differences derived from the adapted time series are typically larger than the trend differences
obtained from the full time series on an absolute scale. A factor 2 is a common feature. In the MLS data, significant trend



differences between the full and the adapted time series are observed around 40 hPa. These differences should to be kept in
mind when comparing results for Boulder and the zonal mean on the shorter time scales.
In addition, we analysed trend differences between Boulder and the zonal means for a number of latitude bands. This aimed
to investigate how representative the Boulder trends are for a more global scale. For the time period from 1987 to 2010 the
simulations indicate trend differences within the interval from $-0.04\,\mathrm{ppmv \cdot decade^{-1}}$ to $0.06\,\mathrm{ppmv \cdot decade^{-1}}$. The largest
differences occur when the Boulder trends are compared to those for the zonal mean of the latitude band between $60°$S and
$60°$N. Based on these results, the Boulder trends should be quite representative (or a reasonable first guess) for the trends on
more global scales during this time period. The caveats regarding missing variability or sub-grid processes in the simulations
apply here as well. For shorter time periods, as covered by the individual satellite data sets, the representativeness gets smaller
in general.
From our analysis it appears that a continued search for the reasons of the trend discrepancies between the FPH observations
at Boulder and the merged satellite set is necessary (see list in the Introduction). In addition, even more differences become
apparent. To start with, this considers the simulations themselves. The overall spread among the trend estimates derived from
the different simulations is typically between $0.1\,\mathrm{ppmv \cdot decade^{-1}}$ and $0.2\,\mathrm{ppmv \cdot decade^{-1}}$. This certainly sheds a different
light on the trend discrepancies between the FPH observations and the merged zonal mean satellite data set, if the spread
among different simulations amounts to a significant part of the discrepancies themselves. The reasons for the spread among
the simulations are probably manifold, comprising general model characteristics, the choice of the nudged reanalysis data
(and their quality over time, Fujiwara et al., 2017) or the exact details of the nudging (see Sect. 2.1). Our analysis does not
provide clear hints in a specific direction but leaves room for obvious followup activities. Then, the trend estimates obtained
from the simulations also differ from those derived from the FPH observations and the merged satellite data set (compare
Fig. 1 and Fig. 5). Overall, they are closer to the trend estimates from the merged satellite data set, but consistently larger
by about $0.02\,\mathrm{ppmv \cdot decade^{-1}}$ to $0.15\,\mathrm{ppmv \cdot decade^{-1}}$ depending on simulation and altitude. Compared to the FPH trend
estimates the model results are consistently smaller by about $0.1\,\mathrm{ppmv \cdot decade^{-1}}$ to $0.45\,\mathrm{ppmv \cdot decade^{-1}}$. In many ways
this situation is reminiscent to the results presented by Garcia et al. (2007) that indicated clear trend differences between the
FPH observations, HALOE and an older version of WACCM for the time period between 1992 and 2002.
A way forward is certainly to go away from derived quantities and to look at difference time series of the water vapour
anomalies instead. An example of this is shown in Fig. 8 which considers the difference between the de-seasonalised time
series derived from the FPH observations at Boulder and the Boulder time series from the different simulations and satellite
observations (i.e. FPH minus the other data sets) used in this work at 70 hPa. For the simulations and the MLS data set the
adapted Boulder time series are used while for the HALOE and MIPAS data sets the full time series are employed, as also
indicated in the legend of the figure. For the de-seasonalisation of the FPH time series the same time period is used as for
the model simulations, i.e. from 1985 to 2010 (see Sect. 3.3). For a clearer picture the differences are smoothed with a 1 year
running mean. At least three valid data points during this period are required for a running mean to be considered further. The
differences visible in the figure are also representative for other altitudes, even though some details are different. Those are



also characteristic for differences between FPH time series and the zonal mean time series for the latitude band between 35°N
and 45°N. A number of aspects gain attention:
(1) Before 1986 the differences to FPH observations are predominantly negative (EMAC being the exception), while after-

wards until 2011 they are mostly positive. As the trends in this work are derived by multi-linear regression with a single

trend term, this behaviour is consistent with larger trend estimates for the FPH observations compared to the simulations

for the time period from the late 1980s to 2010.

(2) In addition, we have in the figure included results derived from observations by the SAGE II (Stratospheric Aerosol and Gas

Experiment II) instrument aboard ERBS (Earth Radiation Budget Satellite), based on the retrieval version 7.00 (Damadeo

et al., 2013). These data are also considered in the merged satellite data set and actually define the start of the time series

(Hegglin et al., 2014). The results here consider the zonal mean time series for the latitude band between 35°N and 45°N (as

noted in the legend). While the SAGE II differences to the FPH observations mostly blend with the other data sets there is

pronounced deviation between 1989 and 1991 (afterwards data are screened due to aerosol contamination by the Pinatubo

eruption). During this time period the differences are more negative than for the model simulations. This behaviour is

consistently observed below 30 hPa. Since this is close to the very beginning of the merged time series it has a pronounced

effect on the trend estimates. It provides an explanation why the trend estimates derived from the merged satellite data set

are smaller than those for the simulations considering the time period from the late 1980s to 2010. Overall, this might hint

to a potential issue with the SAGE II data before the Pinatubo eruption. Alternatively, an issue might originate from the

equal weighting of the pre- and post-Pinatubo data in the merged satellite data set. More investigations are required to rule

out any potential issue.

(3) The temporal development of the differences is quite consistent in qualitative terms for the various simulations and obser-

vational data sets. Features like the strong negative differences around 1993/1994, the subsequent increase until 2000, the

relatively constant behaviour from 2001 to 2009 or the decrease starting in 2010 are visible for all simulations and satel-

lite observations. Interestingly, we find a similar behaviour also in difference time series between frost point hygrometer

observations at other stations and the simulations and satellite observations used in this work (not shown). Explicitly, this

applies to the NOAA FPH observations at Lauder (45°S, 169.7°E) and the CFH (cryogenic frost point hygrometer, Vömel

et al., 2007) observations at San Jose (9.9°N, 84.0°W) and Lindenberg, (52.2°N, 14.1°E). In quantitative terms, the consis-

tency of the differences is evidently is less good. The spread among the various data sets is on average 0.25 ppmv and is,

thus, comparable to the differences between the FPH observations and the different simulations and satellite observations

themselves. In particular, between 1980 and 1985 there are huge deviations among the simulations in their differences to

the FPH observations, relating to differences in the initial conditions and the spin up times among the simulations (except

for CLaMS). After this period the average spread decreases to 0.20 ppmv. One caveat is that the time periods for the

de-seasonalisation inevitably vary among the satellite data sets and are different from that used for the FPH observations

(and model simulations). While this affects the absolute differences, tests show that this has no decisive influence on the

overall spread estimate nor the consistency of the temporal development of the differences shown in Fig. 8.



In summary, understanding the differences shown in Fig. 8 and their temporal development, hopefully in combination with the merged satellite data set, should be a focal point of further research on lower stratospheric water vapour. This will inevitably yield better consistency in the trend estimates but also highlight the benefit of combining different data sources, as in situ observations, satellite measurements and modelling efforts.

*Data availability.*

Simulations:

– The data of the EMAC simulation described above will be made available in the Climate and Environmental Retrieval and Archive (CERA) database at the German Climate Computing Centre (DKRZ, website: http://cera-www.dkrz.de/WDCC/ui/Index.jsp). The corresponding digital object identifiers (DOI) will be published on the MESSy consortium website (http://www.messy-interface.org). Alternatively, the data can be obtained on request from Patrick Jöckel (patrick.joeckel@dlr.de).

– The WACCM data can be obtained on request from Doug Kinnison (dkin@ucar.edu).

– The CMAM simulation can be accessed from the following webpage: http://www.cccma.ec.gc.ca/data/cmam/CMAM/index.shtml.

– The CLaMS data can be obtained on request from Felix Ploeger (f.ploeger@fz-juelich.de).

Observations:

– The NOAA FPH data observed at Boulder can be downloaded from the FTP address ftp://ftp.cmdl.noaa.gov/data/ozwv/WaterVapor/Boulder_LEV or alternatively obtained on request from Dale Hurst (dale.hurst@noaa.gov).

– The HALOE data can be accessed on the following website: http://haloe.gats-inc.com/download/index.php.

– The MIPAS data are available on the following website: https://www.imk-asf.kit.edu/english/308.php.

– The MLS data can be downloaded from the following website: https://acdisc.gesdisc.eosdis.nasa.gov/data/Aura_MLS_Level2/ML2H2O.004/.

– The SAGE II data can be accessed from the following website: https://eosweb.larc.nasa.gov/project/sage2/sage2_table.

*Competing interests.* The authors declare that they have no conflict of interest.

*Acknowledgements.* S. Lossow was funded by the DFG Research Unit "Stratospheric Change and its Role for Climate Prediction" (SHARP) under contract STI 210/9-2. The UTLS water vapour measurement record at Boulder, comprised of monthly balloon flights of the NOAA FPH, continues in its 38[th] year thanks to funding from the NOAA Climate Program Office, the NASA Earth Science Division's Upper Atmospheric Composition Observations programme and the US Global Climate Observing System programme. We appreciate the HALOE Science Team and the many members of the HALOE project for producing and characterising the high quality HALOE data set. We would like to thank the European Space Agency (ESA) for making the MIPAS level-1b data set available. MLS data were obtained from the NASA



Goddard Earth Sciences and Information Center. Work at the Jet Propulsion Laboratory, California Institute of Technology, was done under
contract with the National Aeronautics and Space Administration. The EMAC simulations have been performed at the German Climate
Computing Centre (DKRZ) through support from the Bundesministerium für Bildung und Forschung (BMBF). DKRZ and its scientific
steering committee are gratefully acknowledged for providing the high performance computing (HPC) and data archiving resources for
ESCiMo (Earth System Chemistry integrated Modelling) consortial project. We thank Ted Shepherd for valuable comments on an early
version of this manuscript.
We acknowledge support by Deutsche Forschungsgemeinschaft and Open Access Publishing Fund of Karlsruhe Institute of Technology.





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





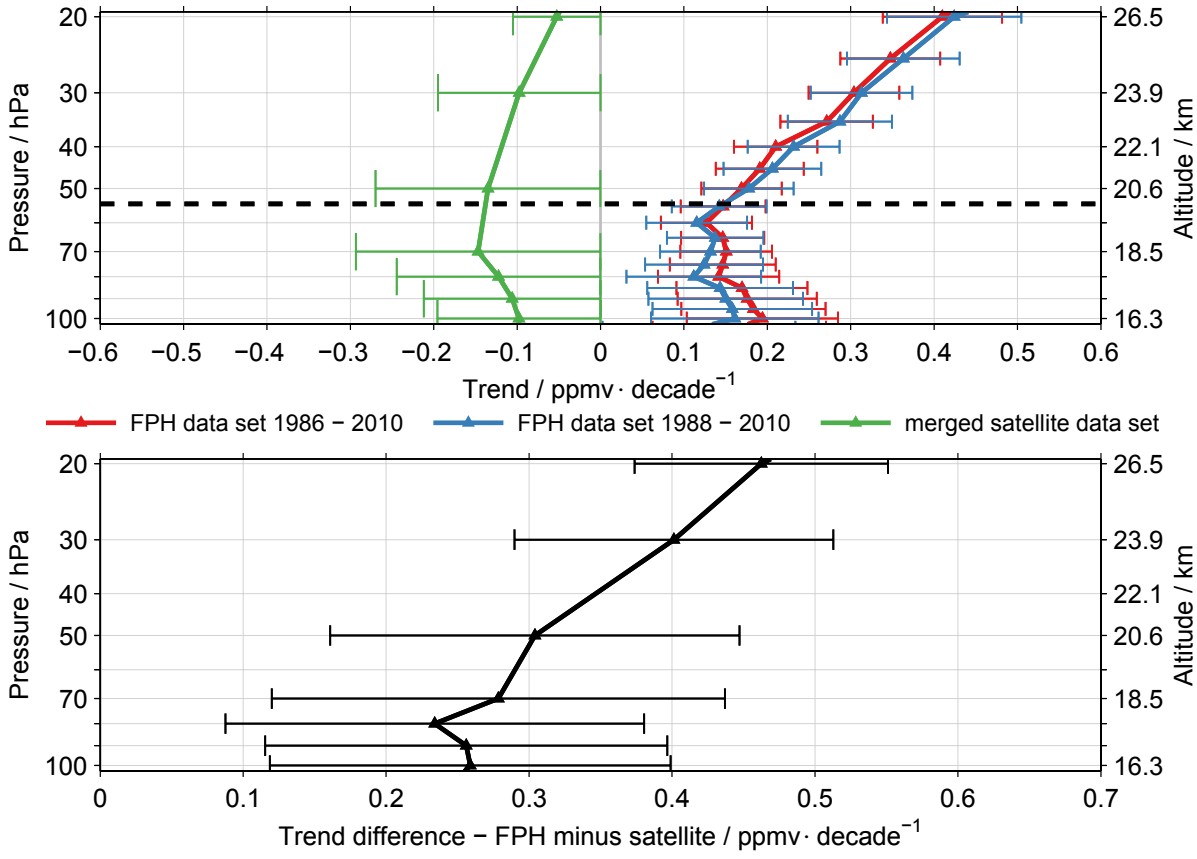

**Figure 1.** Upper panel: In green the approximated trend estimates derived from the merged satellite data set for the the latitude band between 35°N and 45°N are shown. See text for more details. Below the dashed line these estimates consider the time period from 1988 to 2010, above they are for the time period from 1986 to 2010. In red and blue the corresponding trend estimates derived from the FPH observations at Boulder are shown. The error bars represent the $2\sigma$ uncertainty. The right axis provides an approximation of the altitude in geometrical terms. This information is derived from the MIPAS data. Lower panel: Difference between the trend estimates derived from the FPH observations and merged satellite data set.



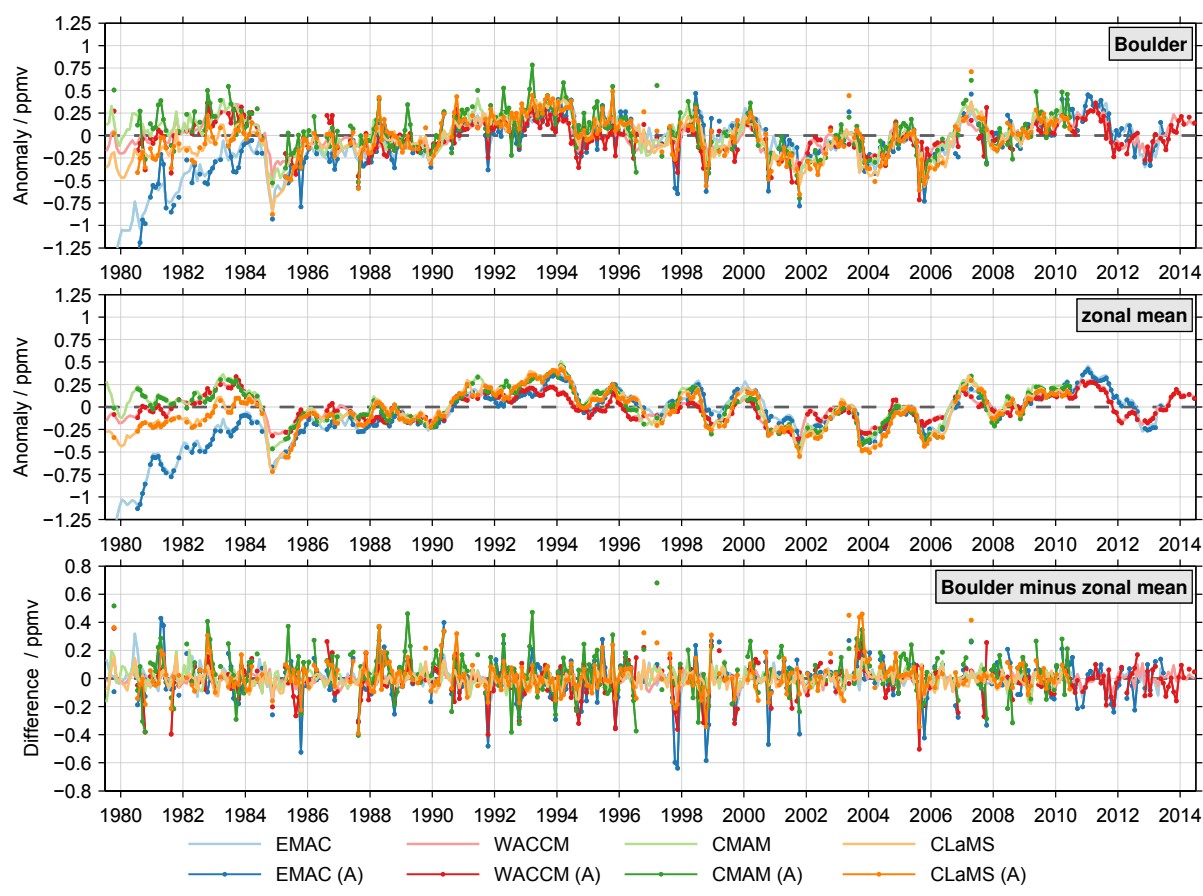

**Figure 2.** De-seasonalised time series for Boulder (top panel), the $35°$N to $45°$N zonal mean (middle panel) and its difference (low panel) for a number of model simulations considering the pressure level of $70$ hPa. Results labelled with the suffix (A) are adapted to the actual FPH observations at Boulder, see text for more details. The time ticks consider the middle of the specified years.





**Figure 3.** Trend estimates for the different model simulations for Boulder (left panels), the zonal mean for the latitude band between 35°N and 45°N (centre panels) and their corresponding differences (right panels). The different rows consider different time period as indicated in the title of the centre panels. Trends and trend differences significant at the $2\sigma$ uncertainty level are marked by triangles.





**Figure 4.** The temporal development of the trend differences between the Boulder and the zonal mean (35°N – 45°N) time series, based on 11 year time intervals. The results are given at the centre of the corresponding time intervals, i.e. in 1995 for the time period between 1990 and 2000. The black lines indicate zero trend differences.



**Figure 5.** Comparison of the trend estimates at Boulder (left panels) and the zonal means for different latitude bands (centre panels) as indicted in the title. As in Fig. 3 the right panels show the difference between the two trends. The comparisons consider the time period between 1987 and 2010. The left panels are all the same and are repeated for convenience.



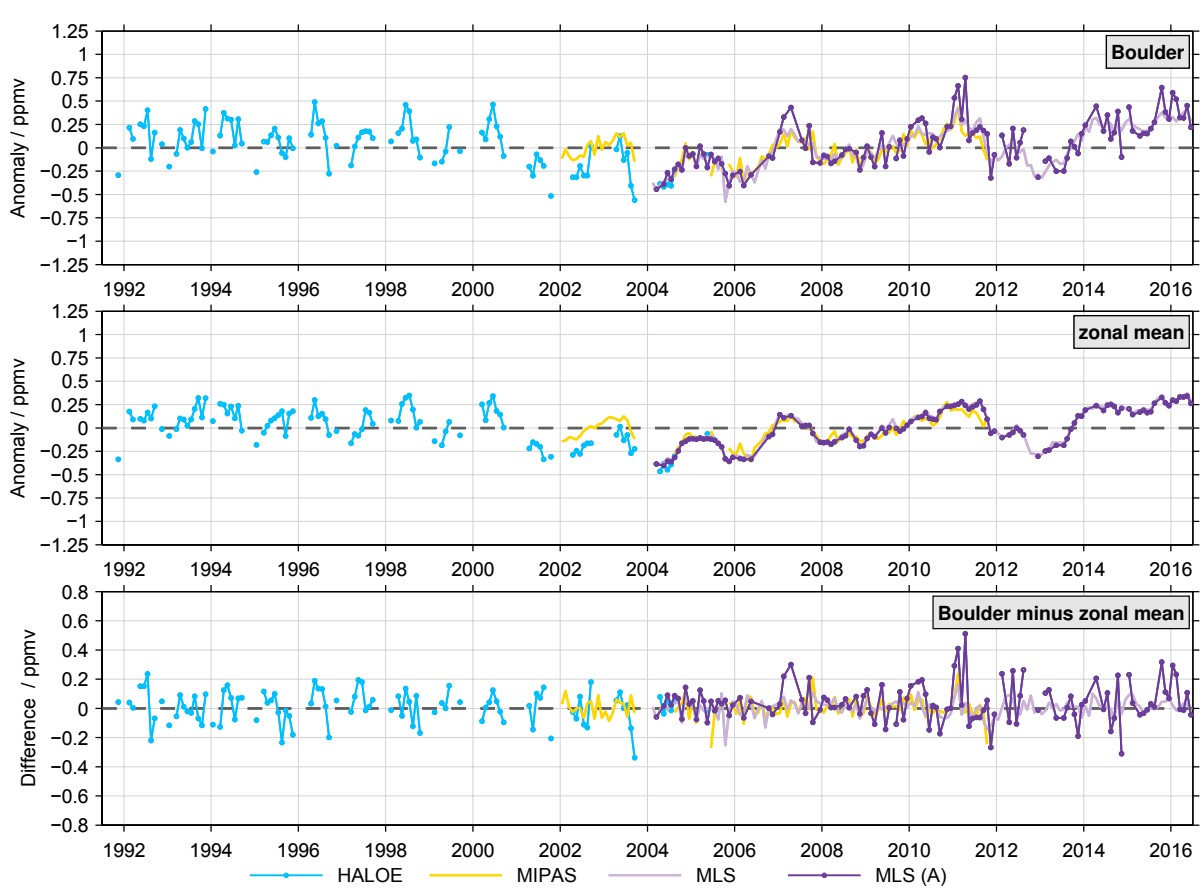

**Figure 6.** As Fig. 2 but here showing several observational results.





**Figure 7.** As Fig. 5 but here again for the observations.




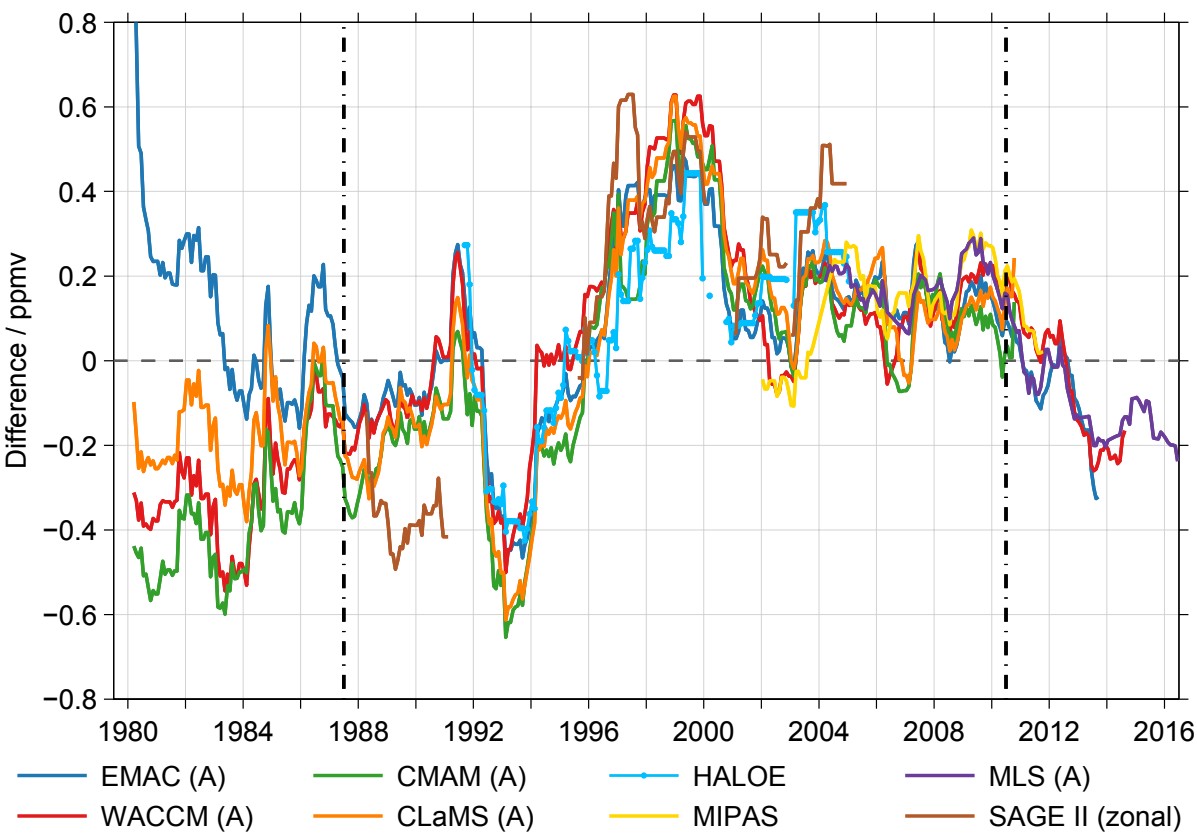

**Figure 8.** Differences between the de-seasonalised time series obtained from the FPH observations and the Boulder time series derived from the different simulations and observational results at 70 hPa. To provide a clearer picture, the differences are smoothed with a 1 year running average. At least three data points are required for a valid running average. The dashed-dotted line indicates the time period covered by the merged satellite data set. The time ticks consider again the middle of the specified years.