# Peer review of "Trend differences in lower stratospheric water vapour between Boulder and the zonal mean and their role in understanding fundamental observational discrepancies"

_Atmospheric Chemistry and Physics, 2017_

## Referee Comment (RC1) · Anonymous Referee #2 · 13 Feb 2018

This is a very good paper and answers (negatively) the important question posed in the title, although given that the question is posed in the title, I do think the answer is surprisingly difficult to find in the text.

My primary concern with this manuscript is that I don't understand exactly how Figure 8, which is an extremely important figure, is produced. In response to a question in the quick review the authors now state: "One caveat is that the time periods for

the de-seasonalisation inevitably vary among the satellite data sets and are different from that used for the FPH observations (and model simulations). While this affects the absolute differences, tests show that this has no decisive influence on the overall spread estimate nor the consistency of the temporal development of the differences shown in Fig. 8." This seems to imply that they have set the average difference equal to zero for each dataset. But if that is the case then, given that Figure 1 shows a trend of ∼0.28 ppmv/decade between satellite and FPH, I do not see how the authors can make the statement that neglecting the fact that the satellite to FPH comparison changes with respect to time period "has no decisive influence on the . . . temporal development of the differences shown". Based on that trend the difference between, e.g., the SAGE vs. FPH differences (average date ∼1996) and the MLS vs. FPH differences (average date ∼2010) must be ∼0.4 ppmv, which is certainly not negligible on Figure 8. On the other hand, my interpretation that the average difference is equal to zero for each dataset is probably wrong (the MIPAS offset appears to be distinctly positive). In any case, the authors should explain how the offsets are calculated and not (absent a much better explanation) say that it doesn't matter.

Abstract page 2 lines 2-4 "Overall, both the simulations and observations exhibit trend differences between Boulder and the zonal mean. The differences are dependent on altitude and the time period considered." I'm not sure what information these lines add (of course there will be some differences) other than to confuse the reader, especially since the next 2 sentences then say that the differences are "not sufficient to explain the discrepancies".

Figure 1 – The error bars for the merged satellite dataset are very hard to see, but, more importantly, on the positive side they all seem to lie exactly on the zero line. Please check to make sure that this is indeed correct, and if it is, please explain why.

Page 7 – Here it says explicitly that: "observations before March 1992 were discarded", yet in several plots data points are shown in 1991. Since what is shown are annual averages this might be mathematically okay, but I would strongly discourage showing

anything before the first data included in the timeseries (at the earliest).

Page 11 – "We focus on the altitude range between 100 hPa and 20 hPa that is typically covered by the FPH observations and in almost all cases completely entirely in the stratosphere (Kunz et al., 2013)." Either "completely" or "entirely" will do, but not both.

Figure 8 – It seems to me that it would helpful to the reader, and would seemingly nicely summarize the main point of the paper, if the authors would add to this figure a line showing Boulder minus zonal mean for any one of the models taken from Figure 2 or 6.

---

## Referee Comment (RC2) · Anonymous Referee #1 · 18 Feb 2018

The present study by Lossow et al. addresses the important question whether the differences in lower stratospheric water vapor trends as derived from the Boulder frost point hygrometer time series and the merged zonal mean satellite data set by Hegglin et al. (2014) are caused by sampling biases. For that purpose, the authors compare water vapour trends at Boulder and for different latitude bands derived from various chemistry-climate models. The same comparison is done for several other satellite data sets. Overall, the analysis indicates that sampling biases are rather not the reason

for the trend discrepancies.

The paper is well written and provides an important contribution to the scientific community. Therefore, I suggest the manuscript for publication in ACP after some, mainly minor modifications.

First of all I have to say that it is a pity that the merged zonal mean satellite data set by Hegglin et al. is not included in the present study, but as I understand the data set is also 3 to 4 years after publication not yet pubicly available, unfortunately. Although Fig. 1 is mainly meant as a motivation to the subsequent analysis, it would be great to see the actual trend estimates for the merged satellite data here, in particular as such figures are often cited and the associated caveats get more and more lost. While extracting the percentage trends from the Hegglin et al. paper is not a problem, the conversion to mixing ratio trends by assuming a fixed reference mixing ratio (same reference value for all altitudes?) is a bit more disturbing.

In some parts the paper is rather lengthy and provides a lot of details, especially in section 4.1. Here the authors provide so many information about simulated water vapour trends at different altitudes, different time periods etc., that it is sometimes difficult to keep focus on the main question of the paper, namely the role of sampling biases in trend estimates. Section 4.1 could as well be part of an evaluation paper on modelled lower stratospheric water vapour trends. For the sake of clarity I would suggest to shorten this section drastically and to focus on one figure that makes the point. Other figures could be moved to a supplement.

By looking at the wide spread in simulated stratospheric water vapour trends I am immediately attempted to ask for explanations for the model spread, but I understand that this would be beyond the scope of the paper (but nevertheless, if there are any ideas, assumptions, etc, it would be great to briefly mention them). However, I am wondering how the choice of the model simulation used as transfer function for the merged zonal mean satellite data set could impact the merged data set? Maybe the

authors could add a short discussion of that issue in section 5.

Overall I can only encourage the authors to continue their research on discrepancies in lower stratospheric water vapour trends among various data sets and to hopefully come up with reliable observational composites, which are key requisites for monitoring changes in atmospheric quantities, but also for model evaluation.

Specific Comments:

- Fig. 1b: Given that the trends from the FPH data and the merged zonal mean satellite data show different signs, plotting the trend difference is a bit confusing to me. This is different to, for example, Fig. 3, since the modelled trends at Boulder and for the zonal mean usually show the same sign.

- Different time periods (e.g. Fig. 3 and 5): The different trend estimates shown in the paper are often based on different time periods, which makes it again sometimes difficult to keep track of the overall picture.

- Fig. 4 and related discussion: The idea behind this figure and the related discussion is not clear to me. I also do not clearly see the link to the shorter observational time series presented in section 4.2. Furthermore, as stated on p 13, l 16/17, the trends shown in Fig. 4 are statistically not significant. Therefore I would recommend to skip this figure.

- Statistical significance: It would be helpful to mention the significance of a trend or difference right away. For example, on p 11, l 4/5 it is stated that "ÂăThe trends derived from the adapted time series yield smaller values as those obtained from the full time series. . .", but later on in the discussion section it is mentioned that these differences are not significant (p 16, l 23/24).

- Why are the FPH trend estimates not included in the various figures for comparison with the model data or the other satellite data sets (e.g. Fig. 3, 5 and 7)?

---

## Author Comment (AC1) · 22 Apr 2018

Replies to the Comments:

The authors thank the reviewers for their insightful comments. In the following, the comments are included in black while our replies are given in blue.

General comments:

The present study by Lossow et al. addresses the important question whether the differences in lower stratospheric water vapor trends as derived from the Boulder frost point hygrometer time series and the merged zonal mean satellite data set by Hegglin et al. (2014) are caused by sampling biases. For that purpose, the authors compare water vapour trends at Boulder and for different latitude bands derived from various chemistry-climate models. The same comparison is done for several other satellite data sets. Overall, the analysis indicates that sampling biases are rather not the reason for the trend discrepancies.

The paper is well written and provides an important contribution to the scientific community. Therefore, I suggest the manuscript for publication in ACP after some, mainly minor modifications.

First of all I have to say that it is a pity that the merged zonal mean satellite data set by Hegglin et al. is not included in the present study, but as I understand the data set is also 3 to 4 years after publication not yet publicly available, unfortunately. Although Fig. 1 is mainly meant as a motivation to the subsequent analysis, it would be great to see the actual trend estimates for the merged satellite data here, in particular as such figures are often cited and the associated caveats get more and more lost. While extracting the percentage trends from the Hegglin et al. paper is not a problem, the conversion to mixing ratio trends by assuming a fixed reference mixing ratio (same reference value for all altitudes?) is a bit more disturbing.

General response #1:

It was our primary wish to include the merged satellite data set. This would have allowed a consistent trend analysis throughout the entire manuscript and also a more detailed look into the SAGE-II data at the beginning of the time series. Accordingly we have asked for it, but got a negative response. In an earlier

version of the manuscript Fig. 1 was not shown and the trend discrepancies were only described in words. However it seemed important to us to show the discrepancies more in detail (in particular the altitude dependence) and that is why we took the effort to digitise Fig. 5a of Hegglin et al. (2014). Of course this caused extra trouble since we had to do an assumption to convert the percentage trends in volume mixing ratio trends, as we arguably do not know how the percentage trends came about. Percentage trends clearly have their value, in this context it seems kind of inappropriate (not to say dangerous). Using a constant value is arguably a crude approach and was primarily chosen for simplicity. Our main motivation was to use a low reference volume mixing ratio so that the trend differences not get larger as they actually are. Being smaller than in reality is not optimal either, but was deemed less harmful. In the new version we use for the conversion a profile based on the MLS observations, to which each satellite data set is finally adjusted in the merging by Hegglin et al. (2014). This profile is based on the average over all MLS observations from August 2004 to December 2010 in the latitude range from 35°N to 45°N. The profile data are provided below on the left (pressure in hPa and volume mixing ratio in ppmv) and it are depicted on the right.

[Figure]

| pressure | vmr |
|----------|-------|
| 100.0 | 4.493 |
| 93.0572 | 4.383 |
| 86.5964 | 4.281 |
| 80.5842 | 4.196 |
| 74.9894 | 4.138 |
| 69.7831 | 4.085 |
| 64.9382 | 4.116 |
| 60.4296 | 4.184 |
| 56.2341 | 4.247 |
| 52.3299 | 4.354 |
| 48.6968 | 4.454 |
| 45.3158 | 4.539 |
| 42.1697 | 4.602 |
| 39.2419 | 4.661 |
| 36.5174 | 4.715 |
| 33.9821 | 4.765 |
| 31.6228 | 4.812 |
| 29.4273 | 4.876 |
| 27.3842 | 4.936 |
| 25.483 | 4.994 |
| 23.7137 | 5.05 |
| 22.0673 | 5.102 |
| 20.5353 | 5.143 |
| 19.1095 | 5.177 |

[Figure]

As a result the volume mixing ratio trends derived from the merged satellite data set get even more negative and the differences to the trend estimates obtained from the FPH observations increase further. The related text has been adapted.

In some parts the paper is rather lengthy and provides a lot of details, especially in section 4.1. Here the authors provide so many information about simulated water vapour trends at different altitudes, different time periods etc., that it is sometimes difficult to keep focus on the main question of the paper, namely the role of sampling biases in trend estimates. Section 4.1 could as well be part of an evaluation paper on modelled lower stratospheric water vapour trends. For the sake of clarity I would suggest to shorten this section drastically and to focus on one figure that makes the point. Other figures could be moved to a supplement.

General response #2:

In an earlier version of the manuscript we only showed the trend differences between Boulder and the zonal mean and additionally considered the time period from 1980 to 2010 (as noted in the text). Given the differences in the model spin-up this time period was not considered any further. However, showing the trends at Boulder and the zonal mean itself was considered important information, in particular as the trends change with period. This makes us very reluctant to cut down on this or to move stuff to a supplement. We have taken away a few sentences, but not really much.

In general, it is undoubtedly true that the manuscript shows more information than needed to answer the perceived key question if a sampling bias could explain the trend discrepancies between the the FPH observations at Boulder and the merged zonal mean satellite data set. This would not needed any analysis of time periods other than from the late 1980s to 2010 nor any involvement of the satellite data in Sect. 4.2. Arguably, this observational discrepancy was the initial motivation, but the choice of the time periods and data sets makes it clear that we wanted to investigate the trend differences between Boulder and the zonal mean in a broader sense. In that regard the title change made in the technical review stage was not a good decision. A more generic title has been now used and the final part of the Introduction has been adapted.

By looking at the wide spread in simulated stratospheric water vapour trends I am immediately attempted to ask for explanations for the model spread, but I understand that this would be beyond the scope of the paper (but nevertheless, if there are any ideas, assumptions, etc, it would be great to briefly mention them). However, I am wondering how the choice of the model simulation used as transfer function for the merged zonal mean satellite data set could impact the merged data set? Maybe the authors could add a short discussion of that issue in section 5.

General response #3:

Reasons for the model differences have been already mentioned in the Discussion section. There we have listed general model characteristics (e.g. convection scheme, wave forcing, parameterisations, etc.), the choice of the re-analysis data set nudged in the simulation or the exact details of nudging (e.g. parameters, top height, relaxation time, etc.).

Different model simulations can influence the merged satellite data set if they differ in their long-term trends. In the merging a bias relative to CMAM is derived (at a given latitude and altitude) for every satellite data set. This bias determination is performed in periods where the individual data sets appear to be problem-free (what the non-problem-free periods mean for the merged data set remains unclear). For CMAM years after 2006 are excluded since there is a problem in the nudged ERA-interim data. Finally, the satellite data sets are adjusted to the MLS data sets using the CMAM derived bias estimates. If the model trends differ the bias estimates for earlier and later satellite data sets will be shifted relatively to each other, reflecting this. This pitfall is discussed by Hegglin et al. (2014) and accordingly the temporal stability of model-measurement differences is assessed. They are claimed to be stable, but to which degree remains unclear. However this is exactly the point that matters, to estimate any additional uncertainty of the merging approach on the trend estimates.

Overall I can only encourage the authors to continue their research on discrepancies in lower stratospheric water vapour trends among various data sets and to hopefully come up with reliable observational composites, which are

key requisites for monitoring changes in atmospheric quantities, but also for model evaluation.

Specific Comments:

Specific comment #1: Fig. 1b: Given that the trends from the FPH data and the merged zonal mean satellite data show different signs, plotting the trend difference is a bit confusing to me. This is different to, for example, Fig. 3, since the modelled trends at Boulder and for the zonal mean usually show the same sign.

Specific response #1:

The difference is simply meant as a summary and a quantification of the trend differences between the FPH and merged satellite data sets. It sets a range how large the sampling bias between Boulder and the zonal mean around Boulder must be to be a valid explanation for the trend discrepancies between the FPH observations and merged satellite data set.

Specific comment #2: Different time periods (e.g. Fig. 3 and 5): The different trend estimates shown in the paper are often based on different time periods, which makes it again sometimes difficult to keep track of the overall picture.

Specific response #2:

See general response #2.

Specific comment #3: Fig. 4 and related discussion: The idea behind this figure and the related discussion is not clear to me. I also do not clearly see the link to the shorter observational time series presented in section 4.2. Furthermore, as stated on p 13, l 16/17, the trends shown in Fig. 4 are statistically not significant. Therefore I would recommend to skip this figure.

Specific response #3:

Figure 4 is part of looking at the trend differences on a broader level. Here, we chose a shorter time period that is more like the time period covered by the

individual satellite data sets. The other motivation was to see how the trend difference varies continuously over time, not only for selected time periods.

None of the trend differences shown in this figure are statistically significant at the 2σ uncertainty level. This is true for most trend differences shown in work (in Figs. 3, 5 and 7 statistical significance is indicated by triangles). This is partly due to approach chosen here. To be consisted with Fig.1 we derived the trends separately for Boulder and zonal mean and finally calculated the differences (Eq. 3). A more elegant way is actually to calculate first the difference time series between Boulder and the zonal mean and then evaluate the trend component of these differences. This approach typically leads to smaller uncertainties in the trend differences. For example, 2% of the trend differences considered in Fig. 3 are statistically significant using this approach. Quantitatively the trend differences are very similar for the two approaches. For the model simulations you hardly see any visual difference. For the satellite observations there are on occasions more obvious differences, in particular for HALOE as it is the sparsest data set. However the overall conclusions of the manuscript are not changed.

Specific comment #4: Statistical significance: It would be helpful to mention the significance of a trend or difference right away. For example, on p 11, l 4/5 it is stated that "The trends derived from the adapted time series yield smaller values as those obtained from the full time series. . .", but later on in the discussion section it is mentioned that these differences are not significant (p 16, l 23/24).

Specific response #4:

The triangles intended to indicate statistical significance are mentioned in the caption of Fig. 3. We mention this now again in the caption of Fig. 5. In general, the comment seems to touch two different things. On page 11, the mentioned text focuses on the EMAC trends at Boulder. Most of those are actually statistically significant, as are a considerable fraction of the trends we show here. On page 16, the text concerns deviations in trend differences derived from the full and adapted time series. We actually do not want to make any statements if those trend differences deviate in a statistical sense, so the word "significance" is not optimal here and has been replaced with "pronounced".

Specific comment #5: Why are the FPH trend estimates not included in the various figures for comparison with the model data or the other satellite data sets (e.g. Fig. 3, 5 and 7)?

Specific response #5:

In an earlier version of the manuscript we actually showed the FPH trends estimates, at least in Figs. 3 and 5. For Fig. 7 it is more difficult since the satellite data sets cover different time periods. There are arguably differences between the FPH estimates and those derived the models and observations (compare Fig.1 and 5), but did not want to put too much emphasis on this. The differences are summarised in words in the Discussion section.

---

## Author Comment (AC2) · 22 Apr 2018

Replies to the Comments:

The authors thank the reviewers for their insightful comments. In the following, the comments are included in black while our replies are given in blue.

General comments:

This is a very good paper and answers (negatively) the important question posed in the title, although given that the question is posed in the title, I do think the answer is surprisingly difficult to find in the text.

My primary concern with this manuscript is that I don't understand exactly how Figure 8, which is an extremely important figure, is produced. In response to a question in the quick review the authors now state: "One caveat is that the time periods for the de-seasonalisation inevitably vary among the satellite data sets and are different from that used for the FPH observations (and model simulations). While this affects the absolute differences, tests show that this has no decisive influence on the overall spread estimate nor the consistency of the temporal development of the differences shown in Fig. 8." This seems to imply that they have set the average difference equal to zero for each dataset. But if that is the case then, given that Figure 1 shows a trend of ~0.28 ppmv/decade between satellite and FPH, I do not see how the authors can make the statement that neglecting the fact that the satellite to FPH comparison changes with respect to time period "has no decisive influence on the ... temporal development of the differences shown". Based on that trend the difference between, e.g., the SAGE vs. FPH differences (average date ~1996) and the MLS vs. FPH differences (average date ~2010) must be ~0.4 ppmv, which is certainly not negligible on Figure 8. On the other hand, my interpretation that the average difference is equal to zero for each dataset is probably wrong (the MIPAS offset appears to be distinctly positive). In any case, the authors should explain how the offsets are calculated and not (absent a much better explanation) say that it doesn't matter.

General response #1:

What Fig. 8 shows can be expressed as follows:

$y_{difference}(t) = \text{running\_average}[\, y_{FPH}(t) - y_{other}(t) \,]$

$y_{FPH}(t)$ is the de-seasonalised time series observed with the FPH instrument at Boulder. $y_{other}(t)$ describes the de-seasonalised time series either from the model simulations or the satellite observations. For EMAC, WACCM, CMAM, CLaMS and MLS we considered what we defined as adapted Boulder time series. For the HALOE and MIPAS instruments the full Boulder time series were used. For SAGE-II the time series for the zonal mean between 35°N and 45°N was implemented. The resulting difference time series was smoothed using a running average of one year, requiring at least three valid data points during this period. If this criterion is not fulfilled the average is discarded. The smoothing is used because otherwise it is difficult to really extract any patterns from the differences.

What has been inconsistent so far was the de-seasonalisation period among the different data sets. For the FPH observations and the model simulations the time period from 1985 to 2010 was used, which is fine. For the satellite observations, however, inevitably the de-seasonalisation period had to be shorter and corresponded to the measurement period of the individual instruments, i.e. from 1992 to 2005 for HALOE, from 2002 to 2012 for MIPAS, from 2004 to 2016 for MLS and from 1986 to 2005 for SAGE-II. The difference time series $y_{difference}(t)$ is of course dependent on the de-seasonalisation periods of the data sets involved and differences in these periods are not optimal. That is why we added the caveat. In the revised version we have now eliminated this inconsistency. For the difference time series $y_{difference}(t)$ the FPH observations now always use the same de-seasonalisation period as the data set they are compared to, i.e. 1985 to 2010 for model simulations (as before), 1992 to 2005 for HALOE, 2002 to 2012 for MIPAS, 2004 to 2016 for MLS and 1988 (at 70 hPa to data just start in this year) to 2005 for SAGE-II.

Part of the caveat on the inconsistent de-seasonalisation periods focused also on the overall spread estimate and the consistency of the temporal development of $y_{difference}(t)$ among the different comparisons. As spread we defined the difference between the maximum and minimum of $y_{difference}(t)$ at given time. The overall spread is the average over all times. In terms of the consistency of the temporal development we referred to the dip in $y_{difference}(t)$ around 1993/1994, the subsequent increase until 2000, the relatively constant behaviour from 2001 to 2009 and so on seen in most comparisons. With the inconsistency of the de-seasonalisation periods now removed we can only

reiterate our caveat statements that this only marginally influenced the overall spread estimate and the consistency of the temporal development.

Abstract page 2 lines 2-4 "Overall, both the simulations and observations exhibit trend differences between Boulder and the zonal mean. The differences are dependent on altitude and the time period considered." I'm not sure what information these lines add (of course there will be some differences) other than to confuse the reader, especially since the next 2 sentences then say that the differences are "not sufficient to explain the discrepancies".

General response #2:

This is simply a summary. Even though this is trivial and presumably the expected behaviour, we think it is still worth to mention this.

Figure 1 – The error bars for the merged satellite dataset are very hard to see, but, more importantly, on the positive side they all seem to lie exactly on the zero line. Please check to make sure that this is indeed correct, and if it is, please explain why.

General response #3:

This is intentional! As described in the text we do not know the exact significance level of the trend estimates derived from merged satellite data set. What we know is that significance level is at least 2 and we assumed this level here for simplicity. What we absolutely wanted to avoid is any overestimation of the significance level. Thus, this conservative approach.

Page 7 – Here it says explicitly that: "observations before March 1992 were discarded", yet in several plots data points are shown in 1991. Since what is shown are annual averages this might be mathematically okay, but I would strongly discourage showing anything before the first data included in the time series (at the earliest).

General response #4:

Nothing is actually shown before that date! Presumably, the confusion arises since the time ticks are placed in the middle of the year, but this is actually noted in the figure captions.

Page 11 – "We focus on the altitude range between 100 hPa and 20 hPa that is typically covered by the FPH observations and in almost all cases completely entirely in the stratosphere (Kunz et al., 2013)." Either "completely" or "entirely" will do, but not both.

General response #5:

Sorry, this is our mistake. The word "entirely" has been removed.

Figure 8 – It seems to me that it would helpful to the reader, and would seemingly nicely summarize the main point of the paper, if the authors would add to this figure a line showing Boulder minus zonal mean for any one of the models taken from Figure 2 or 6.

General response #6:

We have tested this for the different model simulations as shown in the figure below. As expected the results are very similar for the Boulder and zonal mean time series. Because of this we decided not to include any zonal mean data from any of the simulations.

---

## Author Response (AR2)

Dear Stefan!

Please find attached our response to the comments of reviewer #2 and a revised version of the manuscript with the changes colour-coded.

Kind regards,

Stefan on behalf of all co-authors

Replies to the comments of reviewer #2:

The authors thank the reviewer for her/his comments. In the following, the comments are again included in black while our replies are given in blue.

Comment #1: I am happy with almost all of the responses that the authors have provided, but I do not understand the authors continued unwillingness to add a line to Figure 8 showing, for one model, both the local and the zonal differences. As the authors state in their response, "the results are very similar for the Boulder and zonal mean time series". The fact that the results are very similar is a fundamental (if not the fundamental) point of this paper, so why not show this here from the point of view of a model?

Response #1:

We felt that this point had actually already been made by Fig. 2. Of course without smoothing and not involving a third time series, i.e. that from the FPH observations. Also, there were concerns about the text flow regarding Fig. 8 with focuses on Boulder time series, with the SAGE II data being the exception but used for a different purpose. We have reworked this part now and included exemplarily the zonal mean time series from the EMAC simulation.

Comment #2: Also, the title, which now contains the word "and" 3 times is more than a bit awkward. One possibility might be to cut it off after the word "discrepancies".

Response #2:

Since the beginning of this work the title changed probably five or six times. Nothing felt really a hundred percent right. We followed the reviewers advice by ending with "discrepancies" but implemented some smaller changes before. Hopefully, the final title will be "
[revised manuscript text omitted]